https://doi.org/10.1038/s41467-021-27612-x | OPEN

# Intraperitoneal microbial contamination drives post-surgical peritoneal adhesions by mesothelial EGFR-signaling

Joel Zindel [1,2✉], Jonas Mittner [1], Julia Bayer [1], Simon L. April-Monn [3], Andreas Kohler [1], Ysbrand Nusse [2], Michel Dosch[1], Isabel Büchi[1], Daniel Sanchez-Taltavull[1], Heather Dawson[3], Mercedes Gomez de Agüero[1], Kinji Asahina [4,5], Paul Kubes [2], Andrew J. Macpherson[1], Deborah Stroka [1,6] & Daniel Candinas [1,6]

Abdominal surgeries are lifesaving procedures but can be complicated by the formation of peritoneal adhesions, intra-abdominal scars that cause intestinal obstruction, pain, infertility, and significant health costs. Despite this burden, the mechanisms underlying adhesion formation remain unclear and no cure exists. Here, we show that contamination of gut microbes increases post-surgical adhesion formation. Using genetic lineage tracing we show that adhesion myofibroblasts arise from the mesothelium. This transformation is driven by epidermal growth factor receptor (EGFR) signaling. The EGFR ligands amphiregulin and heparin-binding epidermal growth factor, are sufficient to induce these changes. Correspondingly, EGFR inhibition leads to a significant reduction of adhesion formation in mice. Adhesions isolated from human patients are enriched in EGFR positive cells of mesothelial origin and human mesothelium shows an increase of mesothelial EGFR expression during bacterial peritonitis. In conclusion, bacterial contamination drives adhesion formation through mesothelial EGFR signaling. This mechanism may represent a therapeutic target for the prevention of adhesions after intra-abdominal surgery.

[1] Department of Visceral Surgery and Medicine, Inselspital, Bern University Hospital, University of Bern, Bern, Switzerland. [2] Department of Pharmacology and Physiology and Snyder Institute for Chronic Diseases and Department of Microbiology, Immunology & Infectious Diseases, Cumming School of Medicine, University of Calgary, Calgary, AB, Canada. [3] Clinical Pathology Division and Translational Research Unit, Institute of Pathology, University of Bern, Bern, Switzerland. [4] Southern California Research Center for Alcoholic Liver and Pancreatic Diseases and Cirrhosis and Department of Pathology, Keck School of Medicine of the University of Southern California, Los Angeles, CA, USA. [5] Central Research Laboratory, Shiga University of Medical Science, Otsu, Shiga, Japan. [6] These authors contributed equally: Deborah Stroka, Daniel Candinas. ✉email: joel.zindel@dbmr.unibe.ch

The peritoneal body cavity is lined by the peritoneum—a monolayer of mesothelial cells and a sub-mesothelial layer of connective tissue—that allows free movement of intra-abdominal organs. Post-surgical adhesions form when two mesothelial surfaces are attached to each other by connective tissue by a fibrotic reaction, a process that can be initiated by coagulation, aggregation of macrophages, and intercellular adhesions between mesothelial cells[1–3]. The resulting adhesions are defined as irreversible, vascularized fibrotic scars connecting abdominal organs and the abdominal wall at non-anatomic locations, restricting organ movement[4,5]. Post-surgical peritoneal adhesions are a major health burden for patients and health care providers[6]. They are the leading cause of life-threatening intestinal occlusions[7–9] and in the United States alone they are responsible for over 300,000 additional abdominal operations per year with annual costs of several billion dollars[6]. In addition, adhesions frequently lead to chronic post-operative abdominal pain. Currently, the only approved therapies for adhesions are barriers, such as implanted hydrogels, that physically separate internal tissues following surgery. However, the clinical use of hydrogels has not significantly reduced the incidence of adhesion-related disease, and scientific evidence does not support their routine use[10]. Therefore, adhesions are an unresolved clinical challenge which to date lack effective treatment.

A proposed driver of fibrotic conversion is the migration and proliferation of surface mesothelium[11] followed by a mesothelial-to-mesenchymal (MMT) transition[12]. Targeting mesothelial cells reduced adhesion formation in vitro[3] as wells as in vivo in a sterile injury model[11]. However, surgical procedures in the abdominal cavity often require extensive manipulation of the microbe-rich intestines. Therefore, these procedures are not 100% sterile but are often complicated by the contamination of the abdominal cavity with gut microbes. Indeed, some studies link microbial contamination and adhesion formation in humans and rodents[8,13,14]. However, the mechanism how contaminating gut microbes drive adhesion formation remains to be uncovered.

In this work, we ask how microbe-induced inflammation in the peritoneal cavity contributes to adhesion formation. We demonstrate that mesothelial cells are the main source of fibroblast-like cells within adhesions by genetic inducible fate mapping. Using RNA-Sequencing, we show that the activation and trans-differentiation of the mesothelial cell niche are driven by EGFR-signaling, which is significantly upregulated in the presence of gut microbes. These findings are recapitulated in biopsies from human patients. Furthermore, peritoneal adhesions are reduced in a mouse model by targeting EGFR-signaling with the FDA-approved small molecule inhibitor Gefitinib[15].

## Results

**Surgical injury and microbe contamination augment post-surgical peritoneal adhesions**. To investigate the respective effects of sterile injury and microbe contamination on post-surgical adhesion formation, we developed a suitable animal model. First, a standardized surgical injury of the peritoneum was induced by creating a peritoneal button (PB) as previously described[11]. Next, we used a limited cecal ligation and puncture (CLP) to release luminal contents, including microbes into the peritoneal cavity. Together, the PB and CLP components comprise a modular model system with a defined and localized sterile injury due to the PB (Fig. 1a, left panel) and a limited septic insult from the CLP (Fig. 1a, right panel). Both models could be applied individually or in combination (PB + CLP), allowing us to separate the effect of surgical trauma from the effect due to bacterial contamination. The PB + CLP model showed highly reproducible adhesion formation with zero mortality. Adhesions

were evaluated 7 days after surgery using a standardized indexing system (Fig. 1b, c and Table 1). This adhesion index is based on and correlates well with (Fig. S1a) previously published adhesion scores[16–18]. In addition to published scores, that score adhesions for their overall tenacity and vascularization (Table 1), the adhesion index sums up this score from 6 distinct anatomic locations in the mouse peritoneal cavity (Fig. 1b) and therefore reflects the adhesion quantification used in human studies[19].

Taking advantage of our modular adhesion model, we investigated the respective and combined effects of sterile injury and microbial contamination on adhesion formation. Mice underwent either sterile injury alone (PB), microbial contamination alone (CLP), or the combination of sterile injury and microbial contamination (PB + CLP). The combination of microbial contamination and sterile injury led to a significantly higher adhesion index (Fig. 1d) when compared with each insult alone. Next, to distinguish the effect of microbes vs fecal content in adhesion formation, we performed the PB + CLP model in germ-free (GF) mice and in gnotobiotic mice colonized with the stably defined moderately diverse mouse microbiota (sDMDMm2[20]). Both were compared with specific-pathogen-free (SPF) microbiota. Mice were subjected to the PB + CLP model and kept under sterile conditions for one week. We confirmed the hygienic status of the experimental GF and gnotobiotic mice using culture-dependent (aerobic and anaerobic expansion cultures) and culture-independent (Sytox stain) analysis of fecal samples at the end of the experiment (Fig. S1b). GF animals had a significantly reduced adhesion index compared to colonized mice which were like colonized mice receiving injury (PB) only (Fig. 1e). There was no significant difference between the two colonized groups, sDMDMm2 and SPF (Fig. 1e). In addition, adhesions sampled from GF animals showed a decreased collagen content when compared with adhesions sampled from colonized mice (Fig. 1f). Corresponding to a decrease in adhesion formation, GF animals showed a significantly less pronounced increase of pro-inflammatory cytokines after surgery when compared with sDMDMm2 mice (Fig. S1c). Nonetheless, GF animals were able to mount an inflammatory response post-surgery, indicated by a profound influx of inflammatory leukocytes (neutrophils and monocytes) into the peritoneal cavity (Fig. S1d). However, when compared with colonized animals, the infiltration of leukocytes in GF animals consisted of more monocytes and less neutrophils (Fig. S1d). Next, we replaced the CLP with cecal slurry (CS) that was generated from feces of SPF mice. The effect of both, native and heat-inactivated CS on adhesion formation was comparable to CLP (Fig. S1e). Correspondingly, when mice were treated with broad-spectrum antibiotics prior to surgery, no reduction of the adhesion index was observed (Fig. S1f). This does not contradict the GF data where mice had no bacteria prior to surgery. Moreover, none of these regimens completely eradicate bacteria. Interestingly, the microbial contamination (CLP, CS) takes place throughout the entire peritoneum, yet adhesions only occurred locally at the site of injury. The administration of lipopolysaccharide (LPS) also resulted in an increase of adhesions over injury only, but the effect was smaller than that of CLP or CS (Fig. S1e). Taken together, these data suggest that contamination with gut microbes rather than intestinal content, drives the formation of post-surgical adhesions.

**Increase in post-surgical collagen deposition correlates with the activation and proliferative expansion of mesothelial cells**. We next explored what drove collagen deposition after exposure to live gut microbiota. Masson's trichrome stained tissue sections showed consistent collagen deposition in adhesions within 7 days

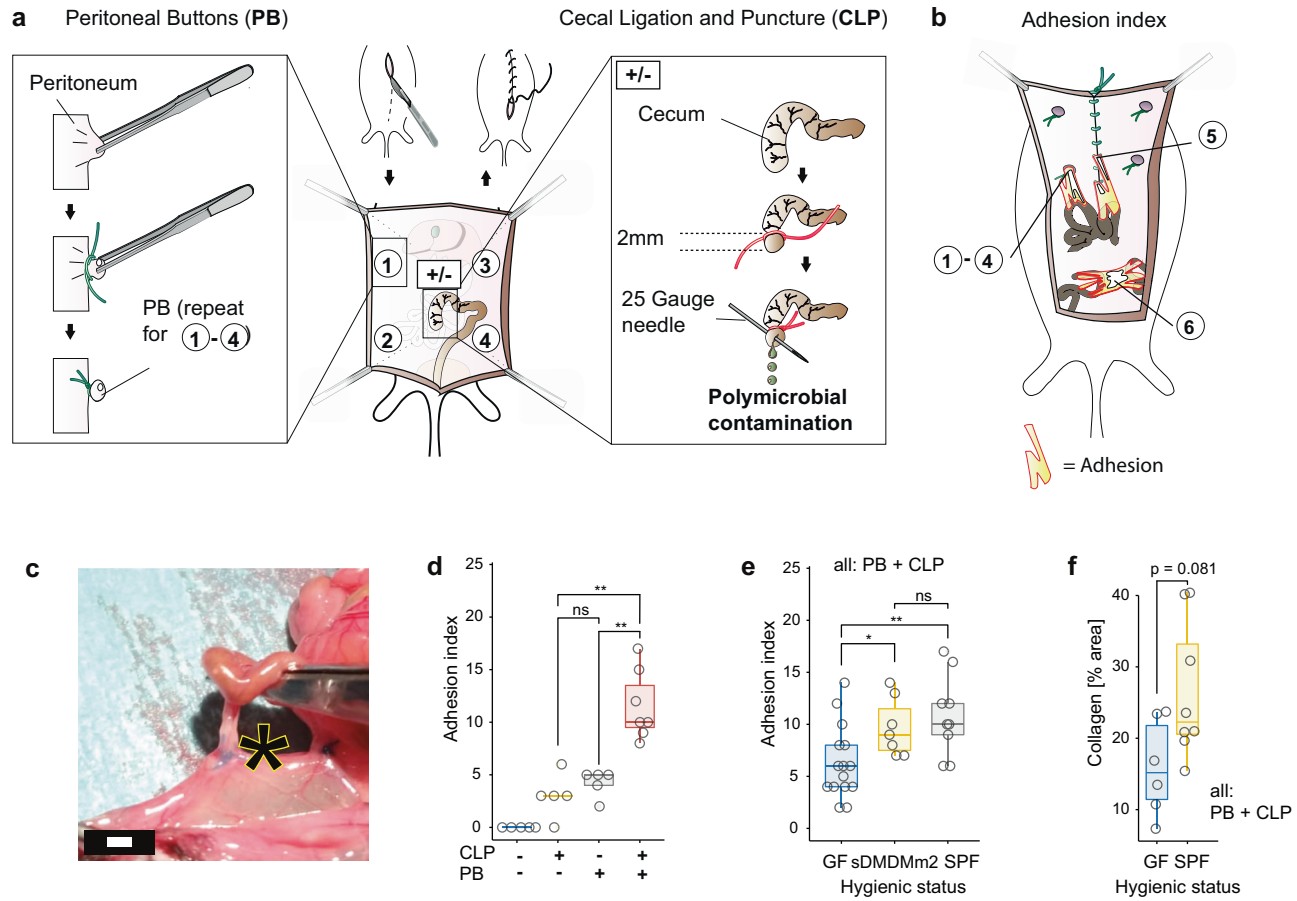

**Fig. 1 Surgical injury and microbe contamination augment post-surgical peritoneal adhesions. a** C57BL/6(J) mice were surgically injured by applying two peritoneal buttons (PB) per side (left panel) in combination with or without microbial contamination through cecal ligation and puncture (CLP) (right panel). **b** Surgical model resulted in the formation of peritoneal adhesions within 7 days post-surgery. Adhesions were scored for tenacity and vascularization at six locations and the sum of these scores is the adhesion index. **c** Representative macroscopic image of an adhesion (asterisk) between small intestine and PB. Scale bar: 1 mm. **d** Adhesion index 7 days post-surgery after CLP, PB, and PB + CLP in SPF mice. Data representative of $n = 5$ for Ctrl, 5 for CLP, 5 for PB and 7 for CLP + CLP independent animals, representing 2 independent experiments. Data are presented as individual values and boxplots (median, first and third quartile). CLP vs PB: $p = 0.39$, CLP vs. CLP + PB: $p = 0.0053$, PB vs CLP + PB: $p = 0.0053$ **e** Adhesion index resulting 7 days post-surgery in germ-free (GF), gnotobiotic (stable defined moderately diverse mouse microbiota, sDMDMm2) and specific-pathogen-free (SPF) mice. All mice underwent PB + CLP surgery. Data pooled from 2 independent experiments representative of $n = 15$ for GF, 7 for sDMDMm2, and 9 for SPF independent animals. Data are presented as individual values and boxplots (median, first and third quartile). GF vs. sDMDMm2: $p = 0.028$, GF vs. SPF: $p = 0.0097$, sDMDMm2 vs. SPF: $p = 0.63$. **f** Collagen quantification (% adhesion area) in GF and SPF mice 7 days after surgery (PB + CLP). Data representative of 6 for GF and 8 for SPF independent animals. Data are presented as individual values and boxplots (median, 25th, and 75th percentile). Wilcoxon test (two-sided) with Holm-Bonferroni correction for multiple testing. *$P < 0.05$, **$P < 0.01$. Source data are provided as a Source Data file.

**Table 1 Peritoneal adhesion index.**

| Grade | Description | Explanation |
|---|---|---|
| 0 | None | PB is free and covered with mesothelium |
| 1 | Flimsy | Separates spontaneously when opening the peritoneal cavity |
| 2 | Dense | Separates bluntly, without bleeding |
| 3 | Fibrotic/Vascularized | Needs sharp dissection, visible vascularization, bleeding occurs upon dissection |
| 4 | Complete | PB is completely covered by adhesion, dissection results in organ damage |

*PB* peritoneal button.

post-surgery (Fig. 2a). Interestingly, the areas in proximity (within 1 mm) of surgical injury/adhesions showed an increased thickness of the sub-mesothelial collagen layer (Fig. 2c, d) when compared with distant (>1 mm) regions (Fig. 2b). The persistence of collagen secreting alpha smooth muscle actin (α-SMA) positive myofibroblasts has been considered a hallmark of fibrotic changes

in wound healing as well as in many fibrotic diseases[21–23]. Therefore, we next probed the question of the origin of α-SMA positive myofibroblasts in adhesions in our model system. We hypothesized that myofibroblasts were either derived from the mesothelium or alternatively derived from sub-mesothelial fibroblasts[24]. To discern these two possibilities, we used conditional

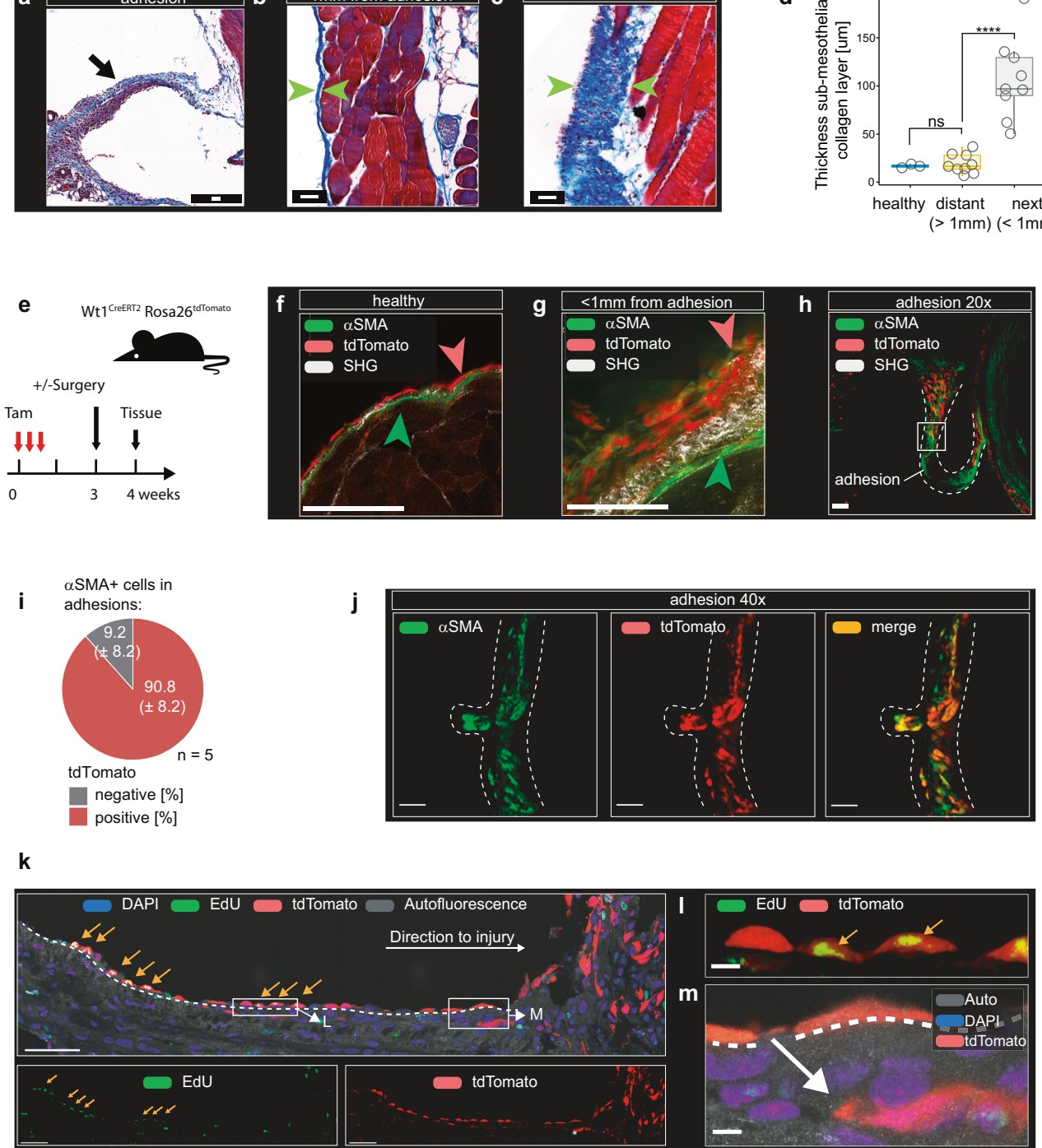

cell lineage tracing. Administration of Tamoxifen to *Wt1*<sup>*CreERT2*</sup> *Rosa26*<sup>*tdTomato*</sup> reporter mice permanently labeled mesothelial cells and their daughter cells with tdTomato (Fig. 2e, f)[25]. In this system, cells derived from sub-mesothelial fibroblasts remain tdTomato negative (Fig. 2f). To appreciate the localization of these cells, we used whole mount microscopy on resected adhesions as previously described[26]. This allowed optical sectioning of adhesions under a confocal microscope with subsequent three-dimensional reconstruction (Fig. S2a). In addition, we used multiphoton excitation to image collagen based on its second harmonic generation (SHG). Under homeostatic conditions, tdTomato positive mesothelial cells did not express α-SMA (Fig. 2f) and were clearly distinct from tdTomato negative, α-SMA positive, sub-mesothelial fibroblasts (Fig. 2f).

However, 7 days after PB + CLP, tdTomato positive mesothelial cells gave rise to α-SMA positive cells (Fig. 2g and Fig. S2b). These tdTomato/α-SMA double-positive cells were rounder when compared with homeostatic mesothelial cells and showed a multi-layered cell growth, compared to baseline mesothelium's monolayer growth (Fig. 2g). Interestingly, mesothelial cells also become α-SMA positive when cultured in vitro (Fig. S2c) and α-SMA positive cells derived from mesothelial cells secreted collagen in vitro (Fig. S2d). In vivo, this phenotypic switch of mesothelial cells was concentrated at the sites of surgical injury (PB) and was associated with an increased thickness of the sub-mesothelial collagen layer, suggesting an increase in collagen production (Fig. 2d). Importantly, when examining α-SMA positive cells within adhesions, we found that many were

**Fig. 2 Increase in post-surgical collagen deposition is correlated with activation and proliferative expansion of mesothelial cells. a–c** Biopsies 7 days after surgery (PB + CLP) stained with Masson's trichrome staining. Biopsies were obtained from adhesions (**a**, arrow), distant regions (**b**) and regions within 1 mm of adhesions in animals that underwent surgery (**c**). Scale bar (**a–c**): 50 μm. Images (**a–c**) are representative of n = 9 animals examined over 3 independent experiments. **d** Sub-mesothelial collagen layer thickness was quantified in biopsies from distant regions and regions within 1 mm of adhesion. Data represent n = 3 for healthy, 9 for distant and next independent animals examined over one independent experiment. Data are presented as individual values and boxplots (median, first and third quartile). Healthy vs. distant: p = 0.86, distant vs. next: p = 0.0000041. **e** Administration of Tamoxifen (Tam) to *Wt1*[CreERT2] *Rosa26*[tdTomato] reporter mice permanently labeled mesothelial cells and their daughter cells with tdTomato. **f–h** Whole mount immunohistochemistry of biopsies obtained 7 days after surgery (PB + CLP). Green arrows indicate sub-mesothelial fibroblasts, red arrows indicate mesothelial cells. Collagen is visualized by its second harmonic generation (SHG). Scale bar: 50 μm. **i** Quantification of *Wt1*[CreERT2] *Rosa26*[tdTomato] positive and negative fraction in alpha smooth muscle actin (α-SMA) positive cells within adhesions. **j** Magnification of adhesion shown in (**h**). Scale bar: 50 μm. **k** 5-ethynyl-2′-deoxyuridine (EdU) was administered twice in the combined injury + CLP model as well as in unoperated control animals during the first twenty-four hours post-surgery. Frozen section immunohistochemistry of biopsies obtained 7 days after surgery (PB + CLP). Yellow arrows indicate proliferating mesothelial cells. Dashed white line indicates basement membrane. Arrow indicates a mesothelial cell crossing the basement membrane. Scale bar: 50um. **l**, **m** magnification (×60) of areas indicated in (**k**). Wilcoxon test (two-sided) with Holm-Bonferroni correction for multiple-testing. ****P < 0.0001, n.s. P ≥ 0.05 n = 5 (**f–j**) and n = 3 (**k**), representative of two independent experiments. Source data are provided as a Source Data file.

tdTomato positive, demonstrating that they were derived from mesothelial cells (Fig. 2h, j). Automated image analysis showed that about 90% of α-SMA positive cells were derived from the mesothelium (Fig. 2i). The capacity of mesothelial cells for mesenchymal transition has been described before and referred to as MMT[12,25,27,28].

In addition to a phenotypic change, the mesothelial cell niche showed a significant proliferative expansion. To show proliferation 5-ethynyl-2-deoxyuridin (EdU) was administered during the first 24 h post-surgery (PB + CLP model) (Fig. 2k and Fig. S2e). Under baseline conditions, very few podoplanin positive mesothelial cells were EdU positive (Fig. S2f). However, post-surgery, the proportion of EdU positive nuclei increased within the mesothelium (Fig. 2k and Fig. S2g). EdU positive cells were often grouped together, distinguishing them as proliferative islands (Fig. 2l and Fig. S2g). These proliferative islands became larger and more confluent near the site of surgical injury (PB), which is also where adhesion formation occurred most frequently (Fig. S2h). At sites of injury, tdTomato positive cells also appeared to lose contact with the basement membrane (Fig. 2k) and to infiltrate the connective tissue (Fig. 2k, m). In summary, our data indicate that α-SMA positive myofibroblasts within adhesions arise from the mesothelial niche which undergoes a proliferative expansion and mesenchymal transition.

**Mesothelial cells undergo a profound transcriptional change post-surgery.** Next, we asked what cell signaling pathways were significantly altered in mesothelial cells after challenging the peritoneal compartment with sterile injury and microbial contamination. We performed RNA-sequencing analysis of mesothelial cells isolated at different time points post-surgery in the combined PB + CLP model. Mesothelial cells were immunopurified using an anti-glycoprotein M6A (GPM6A) antibody and magnetic beads (Fig. 3a). This isolation process resulted in 98% purity when validated by flow cytometry (Fig. 3b and Fig. S3a) and cytospin (Fig. S3b). The RNA from isolated mesothelial cells was processed for next-generation RNA-sequencing. A multi-dimensional scaling plot of all genes and all samples displayed a clear separation of timepoints (Fig. 3c). We identified a total of 9007 differentially expressed genes (DEG) throughout the time course (Fig. S3c). A gene set enrichment analysis was performed to provide an overview of altered pathways (Fig. 3d). We noted an activation of an inflammatory response, including increased cytokine production and the upregulation of canonical leukocyte migration factors (Fig. 3e). In addition, the inflamed mesothelium underwent profound changes in genes involved in the coagulation cascade (Fig. S3d). Taken together, these transcriptomic changes indicate a mesothelial reaction that might facilitate the infiltration

of leukocytes and the formation of a fibrin clot. The latter has been proposed as a preliminary scaffold necessary for a subsequent fibrotic conversion to an adhesion[8].

More than half of the enriched gene ontology terms were not linked to inflammation. The sequential upregulation of genes needed for ribosome biogenesis (Fig. 3d, Nr. 1) was followed by an increase of mitotic cell cycle genes (Fig. 3d, Nr. 2). Several gene ontology terms that were upregulated involved pro-proliferative and anti-apoptotic signaling (Fig. 3d, Nr. 3) supporting the observed expansion of the mesothelial compartment. Further, several intracellular pathways were activated, such as mitogen-activated protein kinase (MAPK), signal transducer and activator of transcription 5 (STAT5) and phosphoinositide 3-kinase (PI3K) signaling, corresponding to cell activation and changes in adhesion, migration, and protein synthesis. In addition, we noted changes in expression of genes associated with cell-cell and cell-basement membrane adhesion/interaction molecules and other markers canonically associated with MMT (Fig. S4a). Altogether, these transcriptional findings suggest that mesothelial cells switch from their epithelial phenotype to assume a more mobile and potentially mesenchymal program.

**Mesothelial cell activation is driven by receptor tyrosine kinases of the ERBB family.** Next, we questioned what was driving proliferation of the mesothelium. Examination of the gene ontology network node "Proliferation/Activation" pointed to receptor tyrosine kinases (RTK) signaling as potential core driver for the observed changes in mesothelial cells (Fig. 3d). Among all RTKs, *Erbb2* showed the highest differential expression (log2 fold change = 2.1, p < 0.001) when comparing germ-free with SPF mice. Similarly, *Egfr (Erbb1)*, was significantly increased after surgery in SPF mice when compared with germ-free mice (Fig. 4c). In addition, the respective downstream pathways of *Egfr* and *Erbb2* were highly differentially expressed (Fig. 4a, b), including genes of the MAPK pathway (Fig. 4b). We next sought to confirm the upregulation and activation of EGFR in mesothelial cells in response to injury and bacterial contamination at the protein level. Interestingly, 24 h after injury, very few mesothelial cells remained within the peritoneal button injury (Fig. 4d). Therefore, we hypothesized that the increased mesothelial *Egfr* expression (Fig. 4a) must come from mesothelial cells that were isolated from regions adjacent to the injury. Indeed, the mesothelium within a few millimeters of peritoneal injuries showed a large increase of EGFR signaling as indicated by the activated form of EGFR (pEGFR) (Fig. 4e) whereas no pEGFR expression was found in distant mesothelium. Importantly, pEGFR showed a high degree of co-localization with the mesothelial cell marker podoplanin, suggesting that the observed

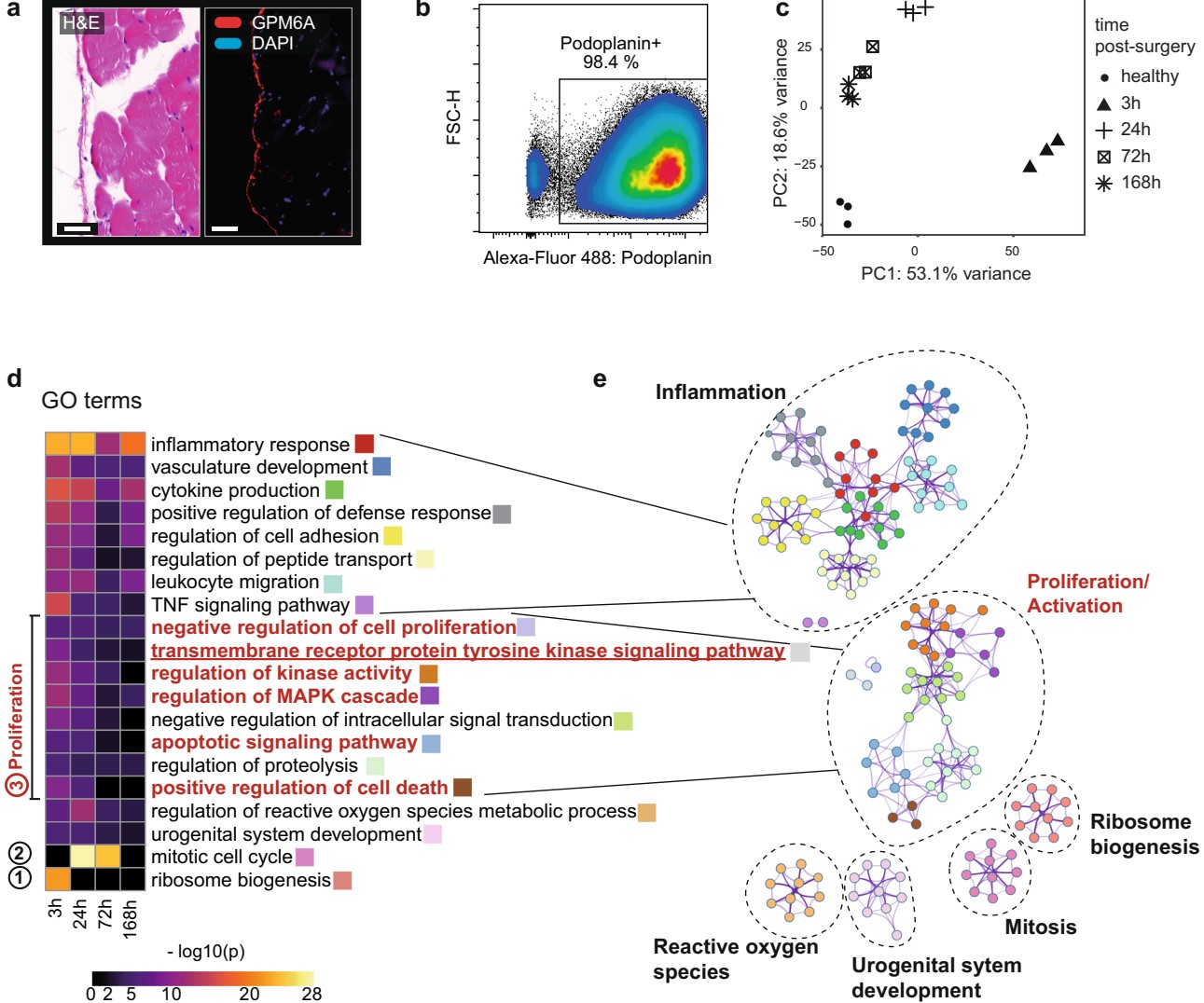

**Fig. 3 Mesothelial cells undergo a profound transcriptional change post-surgery. a** Cryosection of healthy abdominal wall. The mesothelium was stained by anti-glycoprotein M6A (GPM6A) antibody (right). Hematoxylin and eosin (H&E) staining (left). Scale bar: 50 μm. Image is representative of $n = 3$ independent animals and two independent experiments (**b**) In vivo mesothelium digestion purified with anti-GPM6A magnetic beads purity control before undergoing RNA-Seq. **c** Principal components of log transformed read counts. **d** Differentially expressed genes (log2fold change during time-course) with significance ($p < 0.01$) were loaded into Metascape to acquire gene ontology (GO) and KEGG pathway enrichment. Changes in ribosome biogenesis (Nr. 1), mitotic cell cycle (Nr. 2) and pro-proliferative/ anti-apoptotic signaling (Nr. 3) are highlighted. **e** GO network representation of enriched GO terms (color by cluster). $N = 3$ replicates (mice) sequenced per time point. Raw data were deposited at gene expression omnibus (GSE156127).

increase of pEGFR is specific for mesothelial cells. Furthermore, EGFR expression was still elevated 7 days after injury in SPF but not GF mice (Fig. S4b). Taken together, both RNA and protein data suggest that EGFR-signaling is specifically activated in mesothelial cells. This activation occurs in response to injury and is potentiated by microbial contamination.

**EGFR ligands are produced by bone-marrow-derived macrophages and a B-cell subset that are recruited to the wound.** We next asked the question what molecules ligate to EGFR and induce its activation post-surgery. EGFR expression seemed to be predominantly on the basolateral side of the healthy mesothelium (Fig. S5a). We initially hypothesized that surgical disruption of the mesothelial integrity may expose the basolateral receptor to the ligand available in the peritoneal cavity. This hypothesis was further supported by the observation that proliferative mesothelial cells were near the sites of surgical injury (Fig. 2k, Fig. S2f–h). Mesothelial cells produce a certain amount of EGFR ligands in an

autocrine fashion (Fig. S5b). However, EGFR mesothelial ligand transcripts were either unchanged or even decreased in SPF mice when comparing them with GF mice three hours post-surgery (Fig. S5b). We therefore hypothesized that EGFR ligands must be produced in a paracrine fashion by other cells such as peritoneal leukocytes. To investigate the difference in the inflammatory response between mice that underwent PB and PB + CLP we characterized the post-surgical chemotactic signature in the peritoneal cavity lavage fluid using a multiplexed mesoscale cytokine/chemokine screening. Hierarchical clustering of the cytokine/chemokine signature measured in the peritoneal lavage fluid uncovered a distinct proinflammatory neutrophil-recruiting cytokine signature in colonized mice that underwent CLP (Fig. S5c). This proinflammatory signature was well separated from GF mice undergoing CLP and colonized mice receiving only PB without CLP (Fig. S5c). Interestingly, hierarchical clustering revealed that the cytokine signature observed in GF mice with PB + CLP closely resembled that of colonized mice receiving only

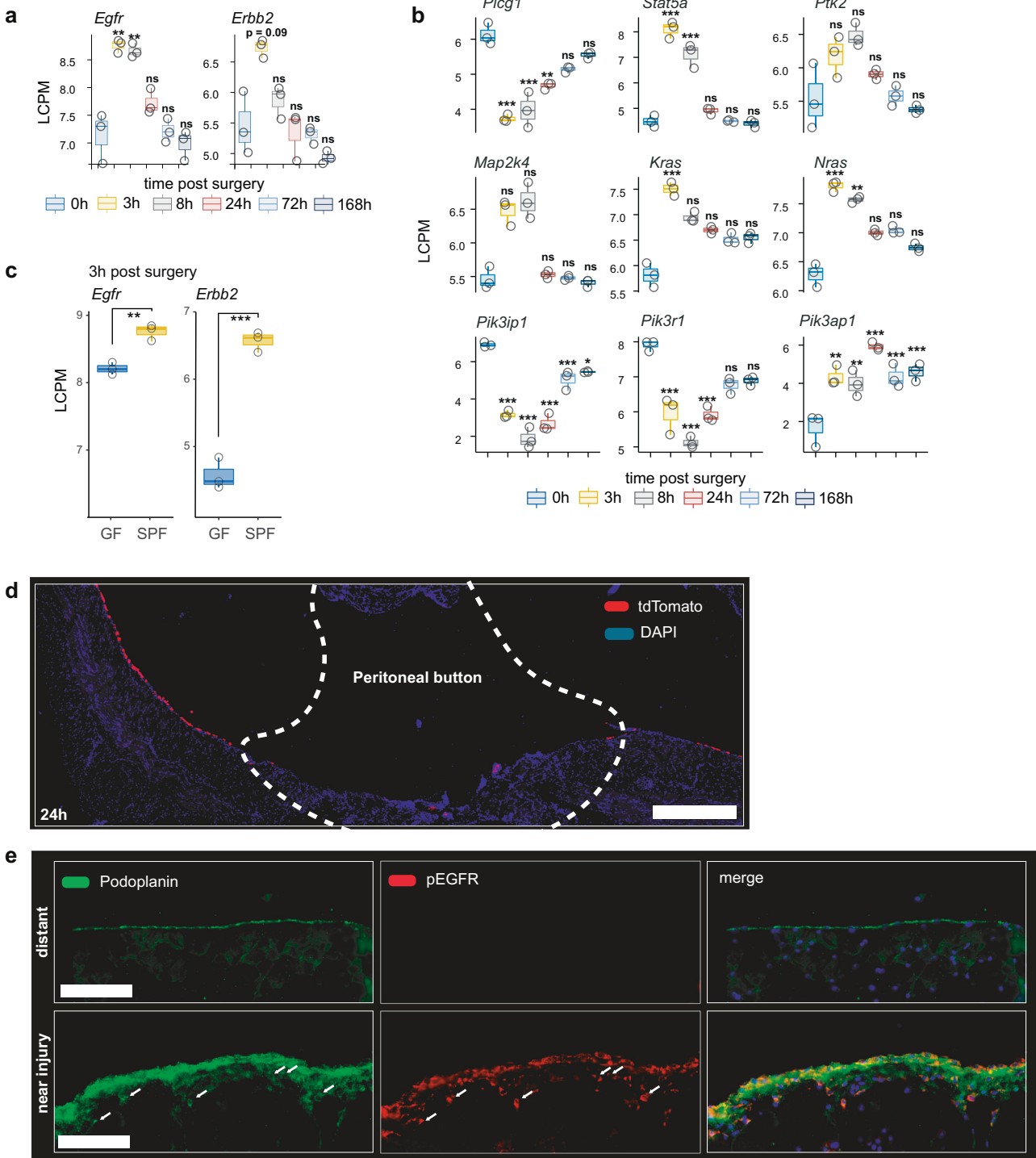

**Fig. 4 Mesothelial cell activation is driven by receptor tyrosine kinases of the ERBB family. a** Epidermal growth factor receptor (*Egfr*) and *Erbb2* were the main genes encoding for receptor kinases differentially expressed (log count per million, LCPM) over time. **b** Genes downstream of *Egfr* and *Erbb2* are differentially expressed as well (inhibitory genes are repressed). **c** Expression of *Egfr* and *Erbb2* genes in mesothelium isolated from germ-free (GF) and specific-pathogen-free (SPF) mice 3 h after sterile injury in combination with bacterial contamination (PB + CLP). Data (**a–c**) is representative of $n = 3$ independent animals examined in one experiment. Data are presented as individual values and boxplots (median, first and third quartile) **d** Peritoneal button, 24 h after injury of *Wt1*^CreERT2 *Rosa26*^tdTomato reporter mice that were treated with Tamoxifen 2 weeks prior to experiment. Scale bar 500 μm. Data represents $n = 3$ animals of 2 independent experiments. **e** Peritoneal biopsies from healthy mice and from mice 3 days after surgery (PB + CLP). Scale bar 50 μm. Data represents $n = 8$ animals of 2 independent experiments. Differential expression testing according to the linear modeling as described in the methods section. *$P < 0.05$, **$P < 0.01$, ***$P < 0.001$, n.s. $P \geq 0.05$. Exact *p*-values are provided in Supplementary Data 1. Raw data were deposited at gene expression omnibus (GSE156127).

the PB without CLP (Fig. S5c). Next, we performed flow cytometric characterization of the leukocyte influx into the peritoneal cavity. Corresponding to the chemokine profile, PB + CLP led to a significantly increased neutrophil recruitment when compared with PB alone (Fig. S5d, e). On the other hand, sterile damage (PB) alone led to an increased influx of monocytes (Fig. S5d, e) whereas the number of macrophages and B-cells was similar in both conditions. The difference between PB and PB + CLP was like the difference between GF and sDMDMm2 observed earlier (Fig. S1c, d). Next, we isolated peritoneal leukocytes 24 h post-surgery and found that peritoneal leukocytes isolated from colonized mice after CLP showed a significantly increased expression (quantitative PCR) of the EGFR ligand encoding genes amphiregulin (*Areg*), epiregulin (*Ereg*), and transforming-growth-factor alpha (*Tgfa*) when compared with peritoneal leukocytes isolated from mice without intraperitoneal microbe challenge (Fig. S5f). These findings suggested that contamination of the peritoneal compartment with live gut microbes leads to an increase in leukocyte recruitment, which produces EGFR ligands in the peritoneal cavity fluid. However, through different assays we were unable to detect EGFR ligands in the peritoneal fluid. We have recently shown, that macrophages can be recruited to peritoneal injuries by a direct route from the peritoneal cavity[2]. Furthermore, a series of recent reports highlights the emerging role immune cell EGFR ligand production in the regulation of inflammation and tissue repair[29–31]. To explore whether EGFR ligands were produced in a paracrine fashion by immune cells that infiltrate the peritoneal injury, we dissociated peritoneal injury biopsies into single-cell suspensions and performed single-cell RNA-Sequencing (Fig. 5a). Manually annotated (Seurat) and automatically annotated (SingleR) clustering confirmed the presence of mesothelial cells (*Krt19*[+], *Gpm6A*[+]) and several distinct populations of CD45[+] immune cells (Fig. 5b, Fig. S6a-c). As expected, the number of mesothelial cells within peritoneal buttons was very small in comparison to the number of infiltrating immune cells (Fig. 4d, Fig. 5c, d). This analysis showed that mesothelial cells were the only cells that expressed *Egfr* (Fig. 5c) but did not express significant amounts of EGFR ligands (Fig. S7a,b). Within injuries, the main ligands with known activity on EGFR homo- and hetero-dimers were heparin-binding epidermal growth factor (*Hbegf*) and *Areg* (Fig. 5d, Fig. S7a,b). The cells expressing the major amounts of *Hbegf* and *Areg* were bone marrow-derived macrophages (Fig. 5d). Interestingly, a small subset of B-cells also expressed *Areg*. This B-cell subset, characterized by the expression of *Ly6d*, *Cd79a*, *Ms4a1,* and *E330020D12Rik*, was only present in mice that underwent CLP in addition to injury (PB) (Fig. 5e). Next, we found that both AREG and HB-EGF led to a significant and dose-dependent increase of EGFR phosphorylation (pEGFR) in cultured primary mesothelial cells (Fig. 5f, g). We went on to interrogate what downstream pathways played a role in our model in comparison with the sterile model by Fischer et al., where ERK did not play a role[3]. In our system, the increase of pEGFR in turn activated MAPK/ERK pathway (Fig. 5g) and higher ligand concentrations led to an activation of the PI3K/AKT pathway (Fig. 5g). We observed no activation of the STAT3 pathway even with high ligand doses (Fig. 5g). Epidermal growth factor (EGF), which was not expressed in our scRNA-Seq experiment, resulted in an even stronger effect in vitro when compared with HB-EGF (Fig. S5g). Next, we tested whether EGFR agonists were sufficient to recapitulate the effect of bacterial contamination in our adhesion model. However, neither the injection of recombinant AREG nor recombinant EGF—which showed the strongest effect on mesothelial cells in vitro—were sufficient to increase the adhesion score in mice that underwent injury model (PB) (Fig. S5h).

**Gefitinib, a small molecule inhibitor of EGFR reduces collagen deposition and MMT in vitro and post-surgical adhesion formation in vivo.** Next, we asked whether pharmacological inhibition of EGFR can be exploited to prevent adhesion formation. Gefitinib was used to inhibit the phosphorylation of EGFR (Fig. 6a, b) and Selumetinib and Ly294002 were used to inhibit the downstream kinases mitogen-activated protein kinase kinase (MEK) and PI3K respectively (Fig. 6a, b). In vitro, Gefitinib led to a significant reduction of the collagen production (Fig. 6c) and migration (Fig. 6d–f) of mesothelial cells. Furthermore, Gefitinib was able to inhibit EGFR-induced MMT in cultured primary mesothelial cells (Fig. S8a). In vivo, the daily intraperitoneal administration of 100 mg/kg of Gefitinib[32] resulted in a significant reduction of post-surgical adhesion formation (Fig. 6g). In addition, intraperitoneal treatment with Gefitinib resulted in a significant reduction of tdTomato positive mesothelium derived cells within adhesions (Fig. 6h, i). We also investigated the administration of Gefitinib by oral gavage with either 20 or 100 mg/kg daily or a once weekly dose of 400 mg/kg as previously described[32]. We found that oral application of Gefitinib or intraperitoneal doses of less than 50 mg/kg per day did not significantly reduce the adhesion index (Fig. S8b, c). This would suggest the need for a high local concentration (μM range) to be effective. Using other kinase inhibitors such as the MEK inhibitor Selumetinib and PI3K inhibitor Ly294002 showed that inhibition of the MAPK/ERK but not the PI3K/AKT pathway results in a reduction of postoperative adhesions (Fig. 6g). Taken together, these findings suggest that EGFR signals through the MAPK/ERK pathway potentiate post-surgical adhesion formation.

**Mesothelial EGFR expression of human patients with acute appendicitis is increased.** To confirm the mesothelial upregulation of EGFR in response to bacterial contamination in humans, we retrospectively analyzed formalin-fixed paraffin-embedded tissues of human patients. Patients either underwent elective surgery due to malignancy without known bacterial peritonitis (control group, n = 7) or due to acute appendicitis (n = 11). The demographics of this patient cohort are displayed in Supplementary Table 1. We hypothesized that EGFR expression in mesothelial cells would be higher in acute appendicitis cases due to the bacterial contamination. Indeed, immunohistochemistry revealed a massive upregulation of EGFR in the whole mesothelium of patients with appendicitis (Fig. 7c–e, Fig. S9a) when compared with patients undergoing elective non-contaminated surgery (Fig. 7a, b, e, Fig. S9b). Interestingly, one outlier in the elective surgery group with relatively high mesothelial EGFR expression, proved to be a patient in which a malignant tumor perforated the intestine which arguably led to a bacterial contamination (Fig. 7e). The EGFR signal showed a very high co-localization with cytokeratin and calretinin, epithelial markers that are expressed by mesothelium (Fig. 7a–d). Furthermore, mesothelial cells from patients with acute appendicitis were significantly rounder when compared with mesothelial cells from patients undergoing elective surgery (Fig. 7c, d, f, g). These observations were consistent with our observations in the mouse model and reports in the literature that suggested that the roundness of mesothelial cells correlates with their ability to migrate[33]. The clear correlation between mesothelial roundness and EGFR expression (Fig. 7g) possibly indicates a relationship between EGFR expression and migration in mesothelial cells. In some acute appendicitis patients, the deposition of granulation tissue enriched in EGFR positive cells (Fig. 7h) could be observed. Drawing from our mouse data, we speculate that these cells were of mesothelial origin and were activated allowing them to proliferate and migrate beyond their basement membrane (Fig. 7h).

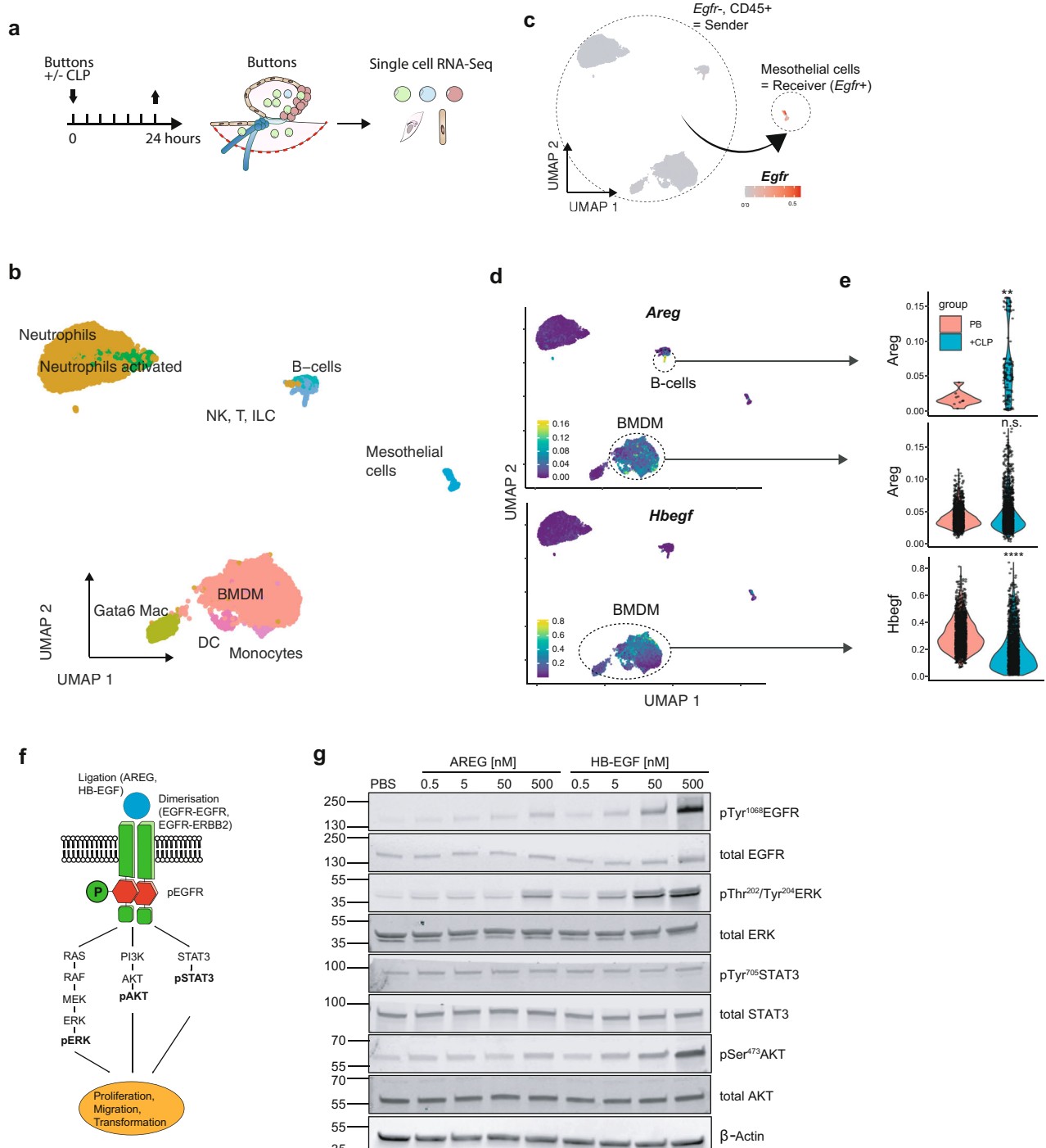

**Fig. 5 EGFR-ligands are produced by bone-marrow derived macrophages and a B-cell subset that are recruited to the wound. a** Illustration of single cell RNA-Sequencing experimental approach. Single cells were isolated from peritoneal buttons (PB) with or without cecal ligation and puncture (CLP). **b–d** UMAP plots colored by cell cluster (**b**), expression of epidermal growth factor receptor (*Egfr*) (**c**), expression of amphiregulin (*Areg*) and heparin-binding epidermal growth factor (*Hbegf*) respectively (**d**). **e** Expression of *Areg* and *Hbegf* by cell cluster and condition. Differential expression testing with FindMarkers function (Seurat). **$P < 0.01$, ****$P < 0.0001$, n.s. $P \geq 0.05$. **f** Schematic illustration of signaling pathways downstream of EGFR. **g** Western blot stained for phospho-EGFR and the respective downstream molecules illustrated in (**f**). Primary mesothelial cells were incubated with amphiregulin (AREG) and heparin-binding epidermal growth factor (HB-EGF) with the indicated doses for 20 min. Single-cell data represents a pooled data set of $n = 4$ mice ($n = 2$ with CLP, $n = 2$ injury alone). The immunoblot shown in (**g**) represents two independent experiments. Raw data were deposited at gene expression omnibus (GSE186658) and Source Data are provided as Source Data File.

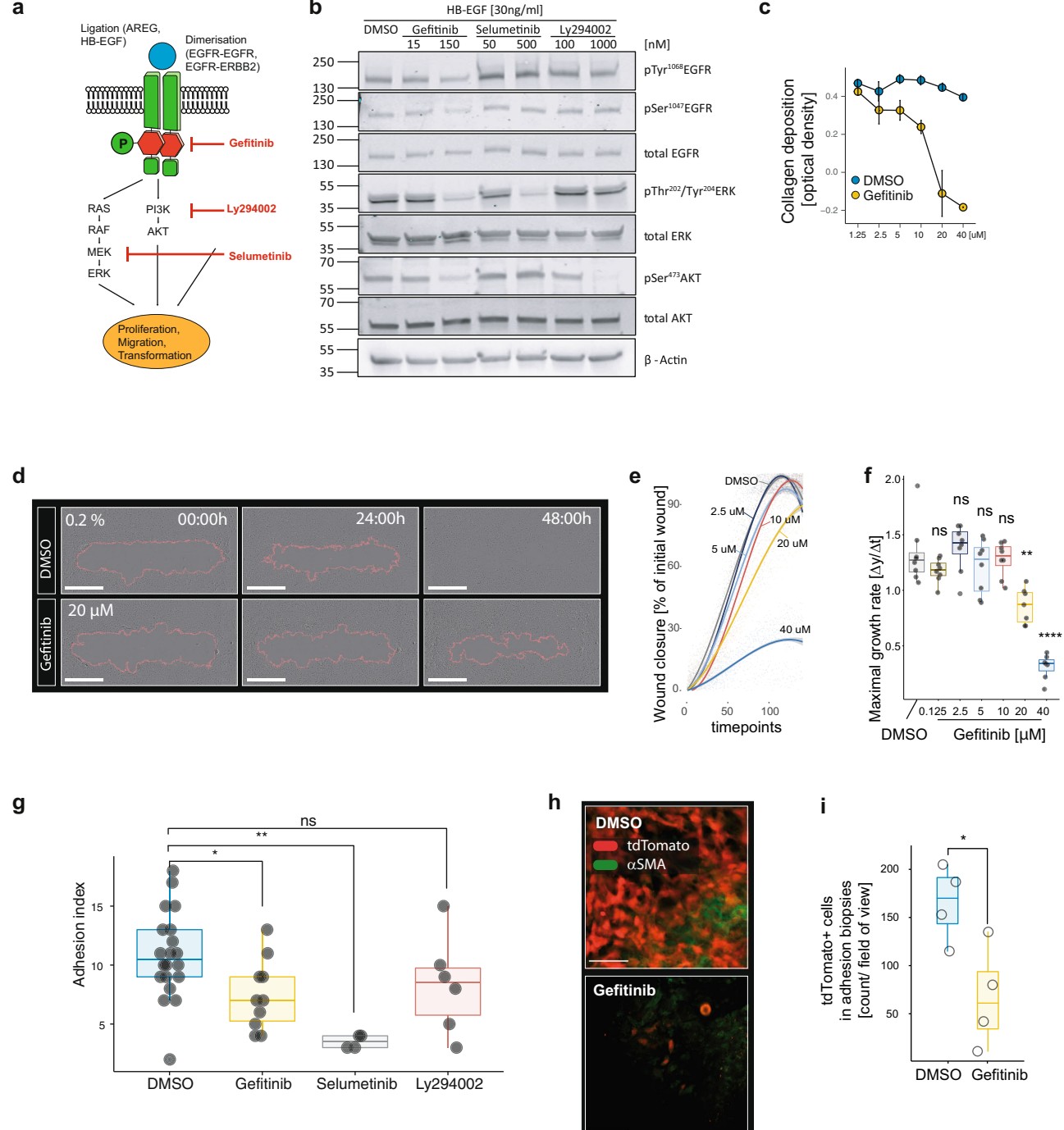

Indeed, when samples were co-stained with the mesothelial markers cytokeratin and calretinin, we found that EGFR positivity was specific for mesothelial cells (Fig. 7h). Cumulatively, these data confirm that EGFR is expressed in activated mesothelial cells also in humans.

**Human patients with fresh adhesions show elevated EGFR-agonist expression.** Next, we wanted to confirm these retrospective observations on a prospective cohort of patients suffering from adhesions. We enrolled $n = 21$ consecutive patients undergoing either elective surgery, emergency surgery for sepsis, or second look emergency surgery. Second look emergency

surgery patients had to be re-operated due to any reason within 14 days after an initial intraabdominal procedure. The presence of adhesions was scored, and patients were grouped into three categories: no adhesions, old adhesions (patient had some adhesions, but they were visibly old and not due to recent surgery) and fresh adhesions either due to abdominal sepsis or recent abdominal surgery. The demographics of this second patient cohort are summarized in Supplementary Table 2. A population of EGFR positive cells was found in biopsies of patients with fresh adhesions (Fig. 8a). These cells also stained positive for mesothelin, suggesting a mesothelial origin of these cells (Fig. 8a). In addition, leukocytes from peritoneal washes of all patients were sampled. Gene expression of EGFR ligands by peritoneal

**Fig. 6 Gefitinib, a small molecule inhibitor of EGFR inhibits adhesion formation in vivo. a** Schematic illustration of signaling pathways downstream of the epidermal growth factor receptor (EGFR). Gefitinib inhibits the kinase domain of EGFR, Ly294002 inhibits phosphoinositide 3-kinase (PI3K), and Selumetinib inhibits mitogen-activated protein kinase kinase (MEK). **b** Western blot stained for phospho-EGFR and the respective downstream molecules illustrated in (**a**). Primary mesothelial cells were isolated and cultured for two passages before they were treated with heparin-binding epidermal growth factor (HB-EGF) and inhibitors for 20 min. **c** Collagen deposited by primary mesothelial within 3 days of culturing. Data represent $n = 3$ technical replicates examined over 2 independent experiments. Data are presented as mean ± standard deviation. **d–f** Primary mesothelial cell cultures were treated with Gefitinib vs. dimethylsulfoxide (DMSO) control and scratch healing was assessed using real-time microscopy (**d**) and automated image analysis (**e**, **f**). Data represent $n = 8$ technical replicates examined over 2 independent experiments. Data are presented as individual values and boxplots (median, first and third quartile). *P*-values by t-test (two-tailed) with Holm Bonferroni correction for multiple testing. Scale bar of (**d**): 1 mm. **g** Adhesion index of mice 7 days after sterile injury (PB) in combination with bacterial contamination (CLP). Gefitinib 100 mg/kg once daily i.p., Selumetinib 50 mg/kg once daily p.o., Ly294002 25 mg/kg once daily i.p., or DMSO 20% once daily i.p. Data represent $n = 20$ for DMSO, 10 for Gefitinib, 4 for Selumetinib, and 6 for Ly294002 independent animals examined and pooled over 3 independent experiments. Data are presented as individual values and boxplots (median, first and third quartile). Wilcoxon test (two-sided) with Holm–Bonferroni correction for multiple-testing. DMSO vs. Gefitinib: $p = 0.017$, DMSO vs. Selumetinib: $p = 0.0058$, DMSO vs. Ly294002: $p = 0.16$. **h** Whole-mount immunohistochemistry of cleared adhesion biopsies 7 days after surgery in *Wt1^CreERT2 Rosa26^tdTomato* mice. TdTomato and alpha smooth muscle actin (α-SMA) positive cells are indicated by red and green respectively. Scale bar 50 µm. **i** Cell count of tdTomato+ cells in adhesion biopsies such as represented in (**h**). Data represent $n = 4$ independent animals per group (averaged over 2 biopsies per mouse, 2–4 fields of view each) examined over one independent experiment. Data are presented as individual values and boxplots (median, first and third quartile). t-test (two-tailed), $p = 0.028$. Statistical difference by. *$P < 0.05$, **$P < 0.01$, ****$P < 0.0001$, n.s. $P \geq 0.05$. i.p. intraperitoneal p.o. per os. Source data are provided as a Source Data file.

leukocytes displayed significantly increased levels of amphiregulin (*AREG*) and epiregulin (*EREG*) in patients with fresh adhesions when compared with both control groups (Fig. 8b, c). Interestingly, epidermal growth factor (*EGF*) was significantly downregulated when compared with controls (Fig. 8d). There was no significant difference in transforming-growth-factor alpha (*TGFA*) (Fig. S10a). In summary, these results replicate two key findings of our mouse model, fresh human adhesions contain EGFR positive cells that are derived from the mesothelium, and human peritoneal leukocytes produce EGFR ligands.

## Discussion

Adhesion formation is driven by a complex interaction of cytokines, coagulation, and growth factors relaying between immune and stromal cells at the site of surgical injury[8]. The duration and severity of the peritoneal inflammatory state is a crucial factor and epidemiologic studies in humans show a correlation between peritonitis, tumor necrosis factor alpha levels, and the severity of adhesions[13,14]. Therefore, immunosuppression has been proposed as potential therapy to attenuate adhesion formation[8]. While this may be potentially rewarding in sterile situations, the use of immunosuppressive drugs seems problematic in cases of bacterial contamination. In fact, in a model of septic peritonitis, while anti-inflammatory and anti-coagulation treatment led to decreased adhesion formation, this treatment significantly increased mortality[34]. This demonstrates that inflammation and coagulative compartmentalization of the peritoneal cavity are important mechanisms of innate immunity that prevent spread of contaminating microbes. However, the resulting adhesions result in considerable morbidity. Therefore, a better understanding of the origin of collagen-producing cells in this fibrotic disease, especially under circumstances of bacterial contamination, may help to find a way to prevent pathologic fibrosis while leaving innate immune function intact.

The capacity of mesothelial cells to undergo a mesothelial to mesenchymal transition has been reported in other diseases[24,25,35,36]. Furthermore, it has been suggested that mesothelial cells are also an important cellular origin of adhesions, as shown with membrane dyes[11,12,37]. More recently, Fischer et al. used an inducible genetic lineage tracing system based on the mesothelial cell marker *Procr* to show that adhesion myofibroblasts arose from mesothelial precursors[3]. Here, we used a *Wt1*-based genetic lineage tracing which aligns with these findings and demonstrates that the vast majority of myofibroblasts within adhesions are derived from the mesothelium,

and not from fibroblasts. Therefore, inhibiting the molecular mechanisms by which mesothelial cells become activated myofibroblasts may provide a means to ameliorate the major source of collagen found in adhesions.

Our data suggest that the proliferation of mesothelial cells is driven by receptor tyrosine kinases of the ERBB family such as EGFR and ERBB2, which is significantly more pronounced in the presence of contaminating gut microbes. We show that EGFR signaling is activated by AREG and HB-EGF, these are EGFR ligands that are derived by leukocytes which infiltrate the wound. EGFR signaling is potentiated by contaminating gut microbes. Firstly, this is due to an increase of EGFR ligand production by immune cells in the peritoneal lavage. Secondly, the mesothelium shows a profound upregulation of the receptor (EGFR) in response to contaminating microbes. The mechanism of this upregulation of EGFR in response to microbial challenge needs yet to be investigated. Furthermore, our data indicate that EGFR is expressed predominantly on the basolateral side of mesothelial cells. This could indicate that the EGFR-ligands produced by immune cells in the peritoneal cavity, only access their receptors at sites of disrupted mesothelial integrity. Indeed, similar mechanisms have been described for repair of injuries to the lung epithelium[38]. Taken together, we provide mechanistic insight into EGFR signaling during post-surgical serosal wound healing and adhesion formation.

Our data suggest that EGFR inhibition may prevent postsurgical formation of adhesions. EGFR inhibition has been reported to prevent generalized peritoneal fibrosis[39], a different disease that shares some pathologic hallmarks with adhesion formation, such as mesothelial origin of collagen secreting myofibroblasts. Here, we show that many key observations in the mouse model were replicated in biopsies of adhesions from human patients. Patients undergoing surgery for acute appendicitis showed significantly elevated EGFR levels in the mesothelium when compared with patients undergoing elective surgery. Additionally, adhesions from patients show the presence of mesothelin/EGFR double-positive cells, suggesting that adhesions may be derived from mesothelial cells. Together, these data suggest that the EGFR-dependent mechanism we identify here is involved in human patients developing adhesions. In conclusion, inhibition of EGFR signaling may represent an avenue for preventing the development of adhesions in patients, by abrogating the expansion and differentiation of mesothelial cells into adhesions. This is particularly interesting because several small-

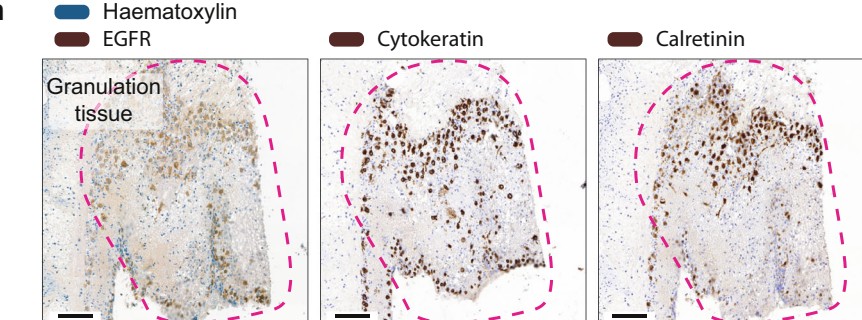

molecule EGFR-inhibitors, like Gefitinib used in this study, have already been approved for the treatment of non-small lung cancer. In our study, high Gefitinib concentrations were necessary to be effective on mesothelial cells in vitro and in vivo. This is a potential limitation to this approach. Further studies, such as retrospective analysis of patients that underwent abdominal surgery while being treated with EGFR-inhibitors as well as prospective studies are warranted to investigate the benefit of EGFR-inhibition in preventing adhesions in human patients.

## Methods

The research conducted complies with all relevant ethical regulations. Human studies and the use of human samples were approved by the Ethical commission of the Canton Bern (project ID: 2017-00573 and 2020-00077). Animal experiments

**Fig. 7 Mesothelial EGFR expression of human patients with bacterial peritonitis is increased. a–d** Biopsies from human patients. Patients either underwent surgery due to acute appendicitis (**c**, **d**) or for elective abdominal surgery such as tumor surgery (**a**, **b**). Formalin fixed and paraffin-embedded samples were stained with anti-human epidermal growth factor receptor (EGFR), anti-human cytokeratin and anti-human calretinin. (**a**) and (**c**) both show an overview. Scale bar: 5 mm. (**b**) and (**d**) show magnifications of (**a**) and (**c**) respectively, Scale bar: 20 μm. **e** Mean EGFR expression was quantified in all patients. Outlier in red color represents an elective-surgery case that turned out to be a perforated tumor with potential bacterial contamination (not excluded for statistical testing). $P = 0.00075$. **f** Automated quantification of mesothelial cell roundness. $P = 0.00013$. **g** Linear regression shows a correlation between mean mesothelial EGFR expression and mean mesothelial roundness ($R^2 = 0.68$, $p < 0.0001$). **h** Area with suspected appendiceal adhesion covered up with granulation tissue. Magnification shows the abundance of EGFR positive cells in granulation tissue. Scale bar: 100 μm. The images shown in (**a–d**) are representative of the quantification shown in (**e–f**). The images shown in (**h**) are representative of the appendicitis group of patients. Data are presented as mean and individual symbols. Data are representative of $n = 7$ for Elective and $n = 11$ for Appendicitis group. Patient demographics according to Supplementary Table 1. Indicated statistical differences in (**e**) and (**f**) by Wilcoxon test (two-sided). R-squared and p-value in (**g**) by linear regression. Source data are provided as a Source Data file. ***$P < 0.001$. OD: optical density.

were carried out in accordance with Swiss federal regulations and approved by the cantonal committee on animal experimentation in Bern Switzerland (BE 18/17 and BE 55/18). The experiments conducted in Canada were conducted in accordance with Canadian legislations and policies and approved by the institutional animal care committee of the University of Calgary in Calgary Canada (AC19-0148 JZ-PA).

**Experimental animals.** Female C57BL/6(J) mice with 8 to 12 weeks of age were purchased from Envigo, Netherlands. Animals were housed in specific-pathogen-free (SPF) conditions with free access to water and food, a 12 h day-night cycle in the central animal facility of the University of Bern, Switzerland. The ambient temperature was 20±2 °C and humidity was kept at 50±10%. Female $Wt1^{CreERT2}$ $Rosa26^{tdTomato}$ reporter mice[25] were housed in SPF conditions with free access to water in the central animal facility of the University of Calgary, Canada. The ambient temperature was 21 °C, and humidity was kept at 32%. Female $Wt1^{CreERT2}$ $Rosa26^{tdTomato}$ reporter were used for experiments at age 10–12 weeks.

**Germ-free and gnotobiotic mice.** Female germ-free C57BL/6(J) mice were derived germ-free as previously described[40] and maintained germ-free in flexible film isolators in the Clean Animal Facility of the University of Bern, Switzerland. Germ-free mice were routinely monitored by culture-dependent (Luria-Bertani broth) and -independent (Gram and DNA-Sytox stains) methods to confirm sterility. Female gnotobiotic C57BL/6(J) mice colonized with stable defined moderately diverse mouse microbiota (sDMDMm2) containing 12 defined bacterial strains were generated[20] and maintained at the Clean Animal Facility of the University of Bern. Gnotobiotic mice were routinely monitored by 16 s rRNA gene sequencing by Ion Torrent PGM system.

**Surgical procedure.** General anesthesia was achieved using isoflurane anesthesia (2% v/v) and analgesia Buprenorphine 0.1 mg/kg body weight (Temgesic®, Indivior, #07680419310018) was administered subcutaneously. The abdomen was then shaved and prepared with alcohol solution. For all surgical models, a 2.5 cm median laparotomy was performed to access the abdominal cavity. Then, lesions were induced to trigger adhesion formation such as peritoneal buttons (PB) and cecal ligation and puncture (CLP). The abdomen was closed using a one-layer running suture (6-0 Prolene®, Ethicon). PB was performed as previously described[41]. In brief, a small portion of the peritoneum is grasped and ligated at its base using a polypropylene suture (4-0 Prolene®, Ethicon), creating a standardized peritoneal button. This is repeated for a total of four buttons, one in each quadrant. A modified sub-lethal CLP was performed. The model was performed as previously described[34]. Different lengths of the cecum were ligated (4 and 2 mm) and punctured once through with a needle of different sizes (18, 21, and 25 Gauge). In the PB + CLP model the lesions of the PB and CLP models were combined. No standard antibiotics prophylaxis was administered. If perioperative antibiotics were given, they were administered 30-60 min prior to surgery by subcutaneous route. The antibiotic substance given were either Ceftriaxone (120 mg/kg, Fresenius Kabi, #61338002), Clindamycine (36 mg/kg, Pfizer, #61898002) or Amoxicillin + Clavulanic acid (200 + 20 mg/kg, Mepha, #56758004).

**Evaluation parameters and tissue collection.** For adhesion scoring and tissue collection, mice were anesthetized by subcutaneous injection of 6 ul/g body weight of a cocktail of Fentanyl, Midazolam, and Medetomidine as previously described[42]. The abdominal wall was accessed using an inverted U-shaped incision and adhesions were scored by two different observers according to the scoring schemes proposed by Nair, Mazuji, and Zuhlke[16–18]. In addition, an advanced scoring scheme called adhesion index was introduced as described in the results section. Blood was collected from the inferior vena cava using a 24 Gauge catheter (BD Insyte-W). Blood was incubated at room temperature for 60-90 min and centrifuged at 2000 × g for 20 min, then the supernatant serum was collected. Collection of peritoneal fluid and peritoneal cells was done as previously described[43]. In brief, the abdominal cavity was flushed with 5 ml of ice-cold phosphate-buffered saline (PBS) which was immediately re-aspirated and snap-frozen in liquid

nitrogen for analysis. Tissue biopsies of peritoneal buttons, peritoneal adhesions and healthy peritoneum control were taken and either snap-frozen or fixed in formalin for 4 h at RT.

**Preparation and administration of small molecule inhibitors.** Gefitinib (Sigma, #SML1657) and Ly294002 (Lucerna Chem, #HY-10108): Stock solutions were prepared by dissolving 100 mg/ml in DMSO. Stock solutions were diluted with saline to reach a final concentration of 20 mg/ml Gefitinib in 20% DMSO. Selumetinib (Lucerna Chem, #HY-50706) was dissolved in 10% DMSO in corn oil. Small molecule inhibitors were administered 2–3 h after the surgery and once daily thereafter. Gefitinib and Ly294002 were administered by intraperitoneal injection and Selumetinib by oral gavage, with the doses as specified in the manuscript and figure legends.

**Cecal slurry (CS) stock preparation.** CS was prepared as previously described[44]. In brief, fecal content from ceca of C57BL/6(J) mice was collected and mixed with sterile water at a ratio of 0.5 ml water to 100 mg of cecal content. The suspension was then filtered consecutively through a 100 μm and 70 μm filter. The filtered solution was then mixed with an equal volume of 30% glycerol in PBS, resulting in a final CS stock solution in 15% glycerol in PBS. The CS stock was aliquoted and stored at −80 °C for later experiments. Heat inactivation of CS was performed by incubating CS stock solution for 20 min at 72 °C. Colony formation assays were performed before and after to confirm heat inactivation.

**Immunohistochemistry.** For immunohistochemistry formalin-fixed paraffin-embedded patient material was cut in 2.5 μm thick serial sections followed by deparaffinization, rehydration, and antigen retrieval using an automated immunostainer (Bond RX, Leica Biosystems, GER). Antigen retrieval was performed for epidermal growth factor receptor (EGFR) with protease for 5 min at 37 °C. EGFR antibody was diluted 1:25 (Supplementary Table 3). Slides were counterstained with hematoxylin. Scans were acquired with an automated slide scanner Panoramic 250 (3DHistech version 3.0.2) at ×40 magnification. Images were analyzed using the QuPath software[45].

**Immunofluorescence.** Formalin-fixed material, incubated overnight in 30% sucrose, was cut at 7 μm in a cryostat (CM3050 S, Leica). After protein blocking (PBS, 5% goat serum), the slides were incubated with the primary antibody with concentrations according to Supplementary Table 3. This was followed by an incubation with a secondary antibody for 1 h at room temperature according to Supplementary Table 3.

**Cytospin.** Mesothelial cells were suspended in PBS containing 3% FCS with primary and secondary antibodies according to Supplementary Table 3. After staining, cells were centrifuged at 800 × g onto a glass slide using Cytospin4® Cytocentrifuge (Thermo Fisher Scientific).

**QuPath image analysis.** Using QuPath software[46] digital-scanned tissue sections of EGFR IHC were first preprocessed in the built-in visual stain editor using default settings for estimation of stain vectors to improve staining quality. In each tissue section, the mesothelium was annotated by a pathologist (H.D.). Using a watershed segmentation method, cells were automatically detected and manually reconfirmed by a pathologist (H.D.) based on histomorphological features including cellular and nuclear shape. A minimum of 800 cells and a total of 77 parameters per cell (including cell perimeter, cell circularity, staining OD etc.) was quantified for each tissue sample. Results were exported as.csv files for statistical analysis in R.

For collagen quantification, sections were stained with Masson Trichrome (Sigma, #HT15-1KT) according to the manufacturer's instruction. Two independent and blinded investigators (JZ, JM) annotated the adhesion area and exported it to ImageJ[46]. The RGB image was split into the respective red, green, and

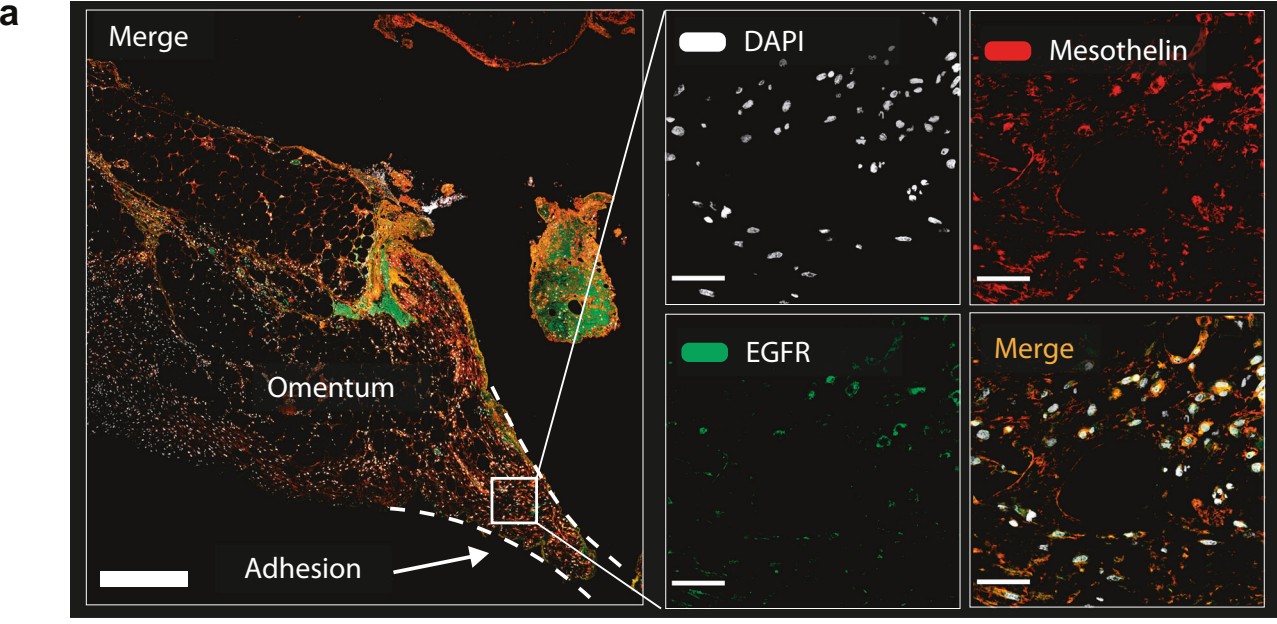

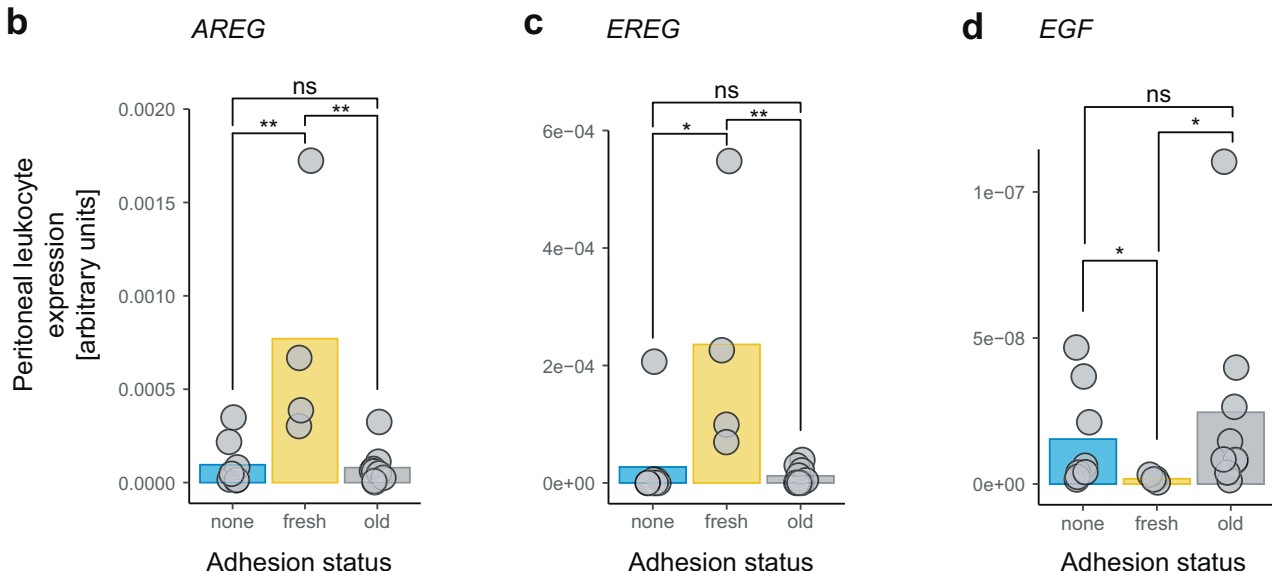

**Fig. 8 Human patients with adhesions have EGFR positive cells of mesothelial origin in them and an elevated EGFR-agonist expression in their peritoneal leukocytes. a** Biopsy from a patient with a fresh adhesion (13 days after initial surgery) stained for mesothelin and epidermal growth factor receptor (EGFR). Image depicts omental tissue that was adherent to a piece of intestine (the latter could not be included in the biopsy). Magnification shows cells that are double positive for mesothelin and EGFR. Scale bar overview: 500 um, Scale bar magnification: 50 μm. Images are representative of $n = 4$ human patients. **b–d** Peritoneal leukocytes were isolated from patients with no, fresh or old adhesions. Expression of EGFR ligand genes was measured by quantitative polymerase chain reaction. (**b**) amphiregulin (*AREG*), none vs. fresh: $p = 0.0081$, none vs. old: $p = 0.96$, fresh vs. old: $p = 0.0056$. **c** epiregulin (*EREG*), none vs. fresh: $p = 0.016$, none vs. old: $p = 0.24$, fresh vs. old: $p = 0.0028$. (**d**) epidermal growth factor (*EGF*), none vs. fresh: $p = 0.028$, none vs. old: $p = 0.48$, fresh vs. old: $p = 0.02$, Data are represented as mean and individual symbols. Data represent $n = 8$ for none, 4 for fresh, and 9 for old individual patients of one independent cohort study. Patient demographics according to Supplementary Table 2. Indicated statistical differences by Wilcoxon test (two-sided) with Holm-Bonferroni correction for multiple-testing. *$P < 0.05$, **$P < 0.01$, n.s. $P \geq 0.05$. Source data are provided as a Source Data file.

blue image components. Collagen formation was quantified by measuring the area percentage of blue channel above a threshold of 120 units (8-bit).

**Isolation of primary mouse peritoneal mesothelial cells.** Mice were anesthetized as described above. A 22 Gauge catheter was inserted into the peritoneal cavity and the peritoneal cavity was flushed three times with 5 ml warm PBS containing 2 mM EDTA. The wash buffer was aspirated completely for each wash and discarded. Then, 5 ml of digestion buffer (0.5% Trypsin-EDTA, ThermoFisher, #15400-054) were injected and incubated for 10 min while the mouse was kept warm under an infrared light. The digestion buffer was aspirated, and the abdominal cavity was flushed three times with 5 ml ice cold PBS containing 2 mM EDTA and 3% FCS to

collect mesothelial cells. Cells were purified as previously described[25]. In brief, cells were washed two times and then incubated with anti-mouse GPM6A (Clone Nr. 321, MBL, #D055-3) for 30 min at 4 °C. After centrifugation the cells were incubated with anti-rat IgG MicroBeads (Miltenyi, #130-048-502) and were purified by MACS Separation columns LS (Milteny, #130-042-401) according to their instructions. Cells were cultured in RPMI 1640 GlutaMAX™ medium (Gibco, #61870044) supplemented with 13% fetal bovine serum, Insulin-Transferrin-Selenium-Sodium Pyruvate (ThermoFisher, #51300044), 20 mM Hepes (Sigma, #H0887), and 100 U/ml Penicillin-Streptomycin (ThermoFisher, #15140122).

**RNA-sequencing of mesothelial cells**. Mesothelial cells at different time-points after surgical induction of peritoneal adhesions using the PB + CLP model were isolated and purified as described above. Cells were kept on ice for a maximum of 30 min. Total RNA was isolated from purified cells by ReliaPrep RNA Miniprep Systems (Promega, #Z6010) according to manufacturer's instructions. RNA quality was assessed using the Bioanalyzer (Agilent 2100 Bioanalyzer) and an RNA 6000 Nano Kit (Agilent Technologies, #5067-1512). Nucleic acid quantification was done using the Qubit RNA Assay (ThermoFisher Scientific, #Q32852). Total RNA was used as input for complementary DNA (cDNA) preparation. Fragments were sequenced using S1 cell flow on a NovaSeq 6000 operated by NovaSeq Control Software (v. 1.5). The reads obtained were trimmed for base call quality and the presence of adapter sequences. Raw fastq files were aligned to the mouse reference genome mm10 using HISAT2[47]. The counts were counted with the featureCounts function of the R package Rsubread. The resulting read counts matrix was analyzed using a standardized Bioconductor workflow with limma, Glimma and edgeR packages in R (23). In brief, raw counts were transformed to counts per million (CPM) after the calculation of library size normalization factors using the edgeR package. Genes not expressing at least 1 CPM in 3 samples were filtered out. For gene differential expression analysis data were transformed and weighted using the voom package. Then a linear model was fitted in the limma package and contrasts were estimated for each gene. A log2-fold change of 1 and $p < 0.01$ were considered as threshold of differentially expressed genes. Differentially expressed genes by limma pairwise comparison were subjected to gene set enrichment analysis using metascape [http://metascape.org](24).

**Single-cell RNA-Sequencing of peritoneal buttons**. Mice C57BL/6(J) underwent surgery to receive injury alone (PB) or in combination with bacterial contamination (PB + CLP). After 24 h mice were euthanized by injection anesthesia and subsequent intracardial perfusion with ice cold PBS with 2 mM EDTA. Peritoneal buttons were excised and digested for 30 min at 37 °C in IMDM (Gibco, #12440061) containing Ca, 25 mM Hepes, 2% FCS, 1 mg/ml Collagenase 1a (Sigma, #C9891-1g) and 0.1 mg/ml DNAseI (Roche, #10104159001). IMDM with 2 mM EDTA was used to stop the reaction and wash the cells. Then, samples were resuspended in 40% Percoll (VWR, #17-0891-01) and pipetted on top of 80% Percoll solution to create a gradient. After centrifugation (650 × g, 20 min), the interface was transferred again and washed in DPBS + 0.04% BSA. Finally, the sample was resuspended in DPBS + 0.04% BSA for library preparation. RNA-seq libraries were prepared from 10000 cells using the Chromium Single Cell 3' Library & Gel Bead Kit v3 (10xGenomics, #PN-1000075). Libraries were prepared according to the manufacturer's protocol. Sequencing was performed on a NovaSeq 6000 S2 flow cell operated by NovaSeq Control Software (v. 1.7). The function cellranger count from Cell Ranger was used to transform the fastq files. The reference genome was the mm10 available at Illumina Cell Ranger webpage. Next, we used the function cellranger mat2csv to generate the UMI matrix.

**scRNA-Seq data analysis**. Data analysis was done following the standard Seurat pipeline[48]. The Seurat objects were created with the function Read10x. Cells expressing less than 200 genes were excluded. Dead cells, identified as cells with more than 10% reads coming from mitochondrial genes, were excluded. Doubles were removed using the doubletFinder function from the DoubletFinder package[49]. The transformation was done using Seurat's SCT transform and the Seurat objects were merged using the merge function. We performed a Principal Component Analysis (PCA) of the Seurat object with the RunPCA function for all the cells. Clustering and dimensionality reduction were performed using FindNeighbors, FindClusters, and RunUMAP (dims = 1:30). Clusters merging and annotation were done manually within the Seurat workflow and unsupervised using the SingleR package[50] for validation. We used the MAGIC package for dropout correction for the gene expression visualization shown[51].

**Western blot**. Protein extraction was performed with RIPA buffer completed with protease inhibitors. Concentration was measured with Bio-Rad microplate protein assay (Biorad, #500-0006). Identical amount of total protein was separated by SDS-PAGE and then transferred on a nitrocellulose membrane (Thermo Fisher, #IB23001) by semi-dry transfer. The membranes were incubated with primary antibodies (Supplementary Table 6) overnight at 4 °C and with secondary antibodies (Supplementary Table 6) for 1 h at room temperature. The proteins of interest were detected using Licor Odyssey infrared scanner operated by Li-cor Odyssey software (2.1.15). For normalization, membranes were incubated with HRP-conjugated β-actin antibody which was detected using enhanced chemiluminescence (WesternBright ECL Spray, Witec, #H K-12049-D50) and the Fusion-

FX7 system operated by the latest firmware (version 1.0.12). All uncropped and unprocessed scans can be found in the Source Data file. All loading controls are displayed in the Source Data file and representative loading controls were chosen for the main text figures.

**Whole-mount tissue staining, clearing, and imaging**. Whole-mount staining and tissue clearing were done as previously described[26]. In brief, mice were perfused with PBS containing 5 mM EDTA. Adhesions/peritoneal buttons were dissected and fixed with 4% PFA/PBS for 2 h at 4 °C. Samples were washed 3 × 30 min in 1% Triton/PBS and then permeabilized and blocked in blocking buffer (1% Triton, 10% FCS, 0.02% sodium azide in PBS). Antibodies were diluted in blocking buffer and incubated for 24–48 h at 4 °C on a rotation device. After washing for 3 × 1 h in blocking buffer and 3 × 10 min in 1% Triton/PBS, the samples were dehydrated using an ethanol series (4 h 50%, 4 h 75%, 2 × 4 h 100%) with a pH of 9.0. Dehydrated samples were incubated in ethyl cinnamate (Sigma, #112372) for 2 h at room temperature and imaged within 2 days with an inverted Leica SP8 2-photon confocal microscope (Leica Microsystems) operated by Leica LAS X software.

**Multiplex cytokine assay (Mesoscale)**. Cytokine measurements were done from mice following PB + CLP model using the Meso Scale Discovery system (MSD, Rockville, Maryland). Serum and peritoneal fluid were collected as described above and stored at −80 °C. The assay was performed according to the manufacturer's instructions. In brief, the MSD system employs a multiplexed immuno-sandwich assay. Each well was prepared with a cocktail of up to 10 specific capture antibodies. Diluted samples and serially diluted standards were pipetted into 2 plates of the customized 19-plex assay. After incubation, cytokine was detected using a cocktail of up to 10 specific, SULFO-TAG-conjugated, detection antibodies. The plate was read in a Meso Scale plate reader and cytokine concentration was calculated from a standard curve, which was fitted for each cytokine using a 4-parameter logistic regression model.

**Flow cytometry**. Suspended cells were isolated from the peritoneal cavity by repeatedly flushing the peritoneal cavity with ice cold PBS containing 3% FCS and 2 mM EDTA (FACS buffer). Cells were filtered through a 40 μm cell strainer (Falcon, #352340) and after centrifugation re-suspended in erythrocyte lysis buffer (Qiagen, #160018730) for 5 min at room temperature. Cells were washed in PBS and stained with fixable viability dye (efluor 506, eBioscience, #65-0866-14) diluted in PBS for 20 min on ice. Single-cell suspensions were incubated with fluorescence-coupled antibodies diluted in FACS buffer according to titration (Supplementary Table 5). Finally, cell data were acquired on a LSR II SORP H271 (BD Biosciences). Flow cytometric analysis was done using FlowJo (Treestar). In all experiments, FSC-H versus FSC-A was used to gate on singlets with dead cells excluded using the fluorescence-coupled fixable viability dye.

**Gene expression analysis**. Tissue samples were snap-frozen in liquid nitrogen. Total RNA was isolated from the tissue by NucleoZOL reagent and following manufacturer's protocol (Macherey-Nagel, #740404.200). RNA concentration and quality were analyzed by spectrophotometer NanoDrop ND-1000 (Thermo Scientific). A total of 500 ng RNA was used for cDNA synthesis by reverse transcription (Omniscript RT Kit 200, Qiagen, #205113). Quantitative PCR was performed using TaqMan gene expression assays (ThermoFisher Scientific) and a real-time PCR cycler (ABI 7900, SDS 2.3 software). Primer and probe sequences were purchased from ThermoFisher Scientific (Supplementary Table 4). Relative changes in mRNA were calculated.

**Human samples**. Both patient cohorts presented in this study were approved by the Ethical commission of the Canton Bern (project ID: 2017-00573 and 2020-00077). All patients gave their informed consent. Participants did not receive any form of compensation. Histology was performed as described above. The isolation of peritoneal leukocytes was performed as previously described[52]. In brief, suction bags were removed after the surgical procedure was finished. The suction fluid was filtered through a 100 um filter and a Ficoll gradient (GE Healthcare, #17-5442-02) was performed by pipetting 10 ml Ficoll under 35 ml peritoneal lavage fluid in a 50 ml falcon tube. Tubes were centrifuged at 800 × g, 4 °C, 20 min, no brake. The interface was collected and transferred to a new tube and after centrifugation resuspended in Erythrocyte lysis buffer (Qiagen, #160018730) for 1 min at room temperature. After one wash with PBS and centrifugation at 800 × g for 5 min, the pellet was lysed for RNA isolation.

**Statistics**. Statistical tests were performed using R[53]. Grouped data were compared using non-parametric tests (Wilcoxon). Multiple testing was corrected using Holm's sequential Bonferroni post-hoc test and $p = 0.05$ was considered the threshold of significance. P-values are graphically represented according to the New England Journal of Medicine style: $p > 0.05$: ns, $p \leq 0.05$: *, $p \leq 0.01$: ** and $p \leq 0.001$: ***.

**Reporting summary**. Further information on research design is available in the Nature Research Reporting Summary linked to this article.

## Data availability

The RNA-Seq data generated in this study have been deposited in the Genome Expression Omnibus (GEO) database under accession code GSE156127. The scRNA-Seq data generated in this study are deposited in the GEO database under accession code GSE186658. The publicly available data (Mus musculus genome assembly, mm10) used in this study are available in the National Center for Biotechnology Information (NCBI) database under accession code GRCm38 [https://www.ncbi.nlm.nih.gov/assembly/GCF_000001635.20/]. The remaining data are available within the Article, Supplementary Information or Source Data file. Source data are provided with this paper.

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

## Acknowledgements

Flow Cytometry & Cell Sorting Core Facility and Live Cell Imaging Core Facility, Department of Biomedical Research, University of Bern. Microscopy was performed with devices supported by the Microscopy Imaging Center (MIC) of the University of Bern. We thank Pamela Nicholson and the Next Generation Sequencing Platform, University of Bern, for their support in planning our scRNA-Seq experiment. They performed RNA quality control assessments, generation of libraries, and sequencing. We thank José A. Galván and the translational research unit of the institute for pathology, they performed the immunohistochemistry on human samples. We thank Riccardo Tombolini for lab management, breeding of mice, and technical support with surgical models. Swiss national science foundation: P1BEP3_181641, J.Z., 310030_179479, A.J.M. European Research Council: ERC advanced grant HHMM-Neonates project no.742195, A.J.M. Natural Sciences and Engineering Research Council: NSERC RGPIN/07191-2019, P.K. Damon Runyon Cancer Research Foundation: Robert Black Fellowship, DRG-2401-20, Y.N.

## Author contributions

J.Z., M.G.D.A., A.J.M., D.S., and D.C. designed the research studies. J.Z., J.M., J.B., Y.N., M.D., and I.B. conducted the experiments. J.Z. and J.M. performed all surgical models. A.K., S.L.A.M., and H.D. acquired and analyzed human data. D.S.T. performed RNA-Seq analysis. K.A. and P.K. provided $Wt1^{CreERT2}$ $Rosa26^{tdTomato}$ reporter mice and the necessary knowledge and infrastructure for their use. A.J.M. and M.G.D.A. provided germ-free animals as well as the knowledge how to use them. J.Z. wrote the manuscript. All authors provided input. D.S. and D.C. were responsible for the overall execution of the study.

## Competing interests

The authors declare no competing interests.
