## [Peer Review File · Nature Communications]

Intraperitoneal microbial contamination drives post-surgical peritoneal adhesions by mesothelial EGFR-signaling.Reviewers' Comments:

Reviewer #1:

Remarks to the Author:

The authors show in their study that in abdominal injuries with bacterial contamination, mesothelial cells express increased levels of EGFR. The EGFR ligands are supplied by immune cells. Inhibition by an EGFR inhibitor reduces the murine adhesions, but does not prevent them. The findings are further supported by patient data.

While the authors provide a solid foundation and evidence for a potential key role of EGFR dependent activation of mesothelial cells, and a potential immune-mesothelial crosstalk, the current version of the study lacks novelty.

EGFR possesses a multitude of downstream effectors and mediators, which affect a variety of cells (e.g. also fibroblasts). But what is the effect on mesothelial cells which leads to adhesion development is understudied here. Furthermore, the authors do not show the activation state of the EGFRs found in mesothelial cells (e.g. phosphorylation Tyr1045). For example, EGFRs also regulate vesicular transport, which in turn plays a key role in the biology of mesothelial cells. The following comments would add value to a revised study:

Major comments:

- Genetic tracing has been shown, even in combination with genetic depletion strategies such as DTA (Fischer et al., 2020), and could be mentioned.

- The transcriptional changes need to be validated at the protein level. It is essential to search for active signaling molecules, as claimed by the authors. Phosphorylated EGFR should be visualized together with mesothelial markers. The cells in the adhesion sites should show increased phospho-EGFR signalling compared to the mesothelial cells in the neighboring sites. This is of essential importance, since EGFR plays a key role in many processes.

Interestingly, ERK signaling does not inhibit adhesion formation in vivo (Fischer et al., 2020 supplementary data). Therefore, the EGFR-dependent downstream effects should be described in more detail and the mode-of-action of EGFR-mediated adhesion formation should be described.

- The authors claim that EGFR-dependent activation of mesothelial cells does not occur through EGF but through ligands such as AREG, EREG or TNF α . Which exact ligands are involved and who is secreting these ligands in mouse and human adhesions. SCRNAseq data and immunostaining could eliminate these points.

- A cell-type specific depletion or cell-type deletion of the ligands should be performed. Is it possible that these ligands activate a distinct EGFR signaling pathway induced by EGF in mesothelial cells?

- Bacterial contamination takes place throughout the entire abdomen, why do adhesions still only occur locally? this could be better explained. Adhesion induced by EGFR alone could therefore be a partial mechanistic story. The authors should therefore investigate how different injury +/- EGFR affects mesothelial cells.

- The treatment regimen with gefitinib raises several questions. 100mg/kg means a daily injection of 2mg/20g mouse. These amounts are unrealistically high for clinical use and could be do to indirect effects, other than on mesothelium. It would therefore be desirable if the authors could show another EGFR inhibitor. Also a titration curve needs to be shown what are the minimal doses that are effective.

-A much deeper analysis of the pathway and mode of action needs to occur for this to be novel.

-Can ectopic ligand administration or EGFR activation induce adhesion formation without injury?

-No direct proof of which immune cells express and secrete ligands are responsible potentially for an immune-mesothelial crosstalk. Depletion schemes or purification and transplantation of immune cells needs to be done.

Minor comments:

- Supplemental figure 6; wrong title

Reviewer #2:

Remarks to the Author:

General comment:

This manuscript deals with an important medical complication, the peritoneal adhesions, which occur very frequently after abdominal surgery. Nowadays, there are no definitive treatments to prevent or to ameliorate the formation of adhesions. In addition, the cellular nature and the molecular mechanisms for adhesion formation are still open questions. In previous publications it has been demonstrated that targeting mesothelial to mesenchymal transition reduced adhesion formation in a sterile injury model. Nevertheless, abdominal surgery is not a sterile trauma and is often complicated by the contamination with gut microbes.

In this study, the authors hypothesized that microbe-induced inflammation contributes to peritoneal adhesion formation. The group employed various mouse models of peritoneal adhesion based on ischemic button, accompanied or not by cecal ligation and puncture and they demonstrate that the release of luminal contents, microbes, into the peritoneal cavity exacerbates the formation of peritoneal adhesions.

By using different experimental approaches, including lineage tracing, immunohistochemical studies and RNA-seq of purified mesothelial cells, the authors demonstrate that mesothelial cells are the main source of myofibroblasts via mesothelial to mesenchymal transition (MMT). Then the authors attempt to connect the MMT process with EGFR-signaling, which is upregulated in the presence of gut microbes. In this context, Gefitinib, a small molecule inhibitor of the EGFR-signaling reduced peritoneal adhesions in the mouse model. Finally, these findings have been partially recapitulated in biopsies from human patients.

Although the results presented are potentially interesting, the experimental approaches are technically sound, and data support the authors conclusions, the study lacks enough mechanistic analysis of the role of EGFR-signaling in both MMT and microbe-induced adhesions.

Specific comments:

1.- No mechanistic studies were performed to define how EGFR ligands induce the mesenchymal conversion of the mesothelial cells. In vitro studies with primary mesothelial cells to analyse the EGFR-mediated MMT and the mechanism of action of Gefitinib would strength author's conclusions.

2.-In the result section, there is a subheading entitled: "Gefitinib, a small molecule inhibitor of EGFR reduces wound healing of mesothelial cells in vitro and post-surgical adhesion formation in vivo". However, these in vitro experiments are not shown.

3.- The authors present a detailed RNA seq and qPCR studies of mesothelial cells isolated by magnetic beads. They have found that a large number of genes are deregulated in the adhesion-inducing conditions at different time points. In figure 4, the authors demonstrate, at the level of gene expression, the activation of the EGFR pathway, at different times, after surgical damage. Likewise,

they show that contamination with microbes induces a hyper-regulation of EGF receptors. However, it is well known that signal transmission through kinase pathways is regulated by post-translational processes including phosphorylation and subcellular localization of kinases. Therefore, a characterization of protein expression and/or post-translational modifications would significantly improve this study.

4.- Surprisingly, they found the up regulation of E-cadherin at early time points post-surgery, in parallel with the up-regulation of the transcriptional repressor Snail (Suppl. Figure 4). In addition, cultured mesothelial cells isolated from peritoneal adhesions showed a round morphology (Suppl. Figure 2). These data are in sharp contrast with what is known in about the MMT process. These apparent discrepancies should be discussed.

5.- Figure 5: A better characterization of the immune/inflammatory response to PB and PB+CLP would reinforce the conclusions. For example, what is the main Th (Th1, Th2, Th17) response in the different conditions? The changes of the inflammatory responses between PB and PB+CLP are only qualitative or both qualitative and quantitative?. What are the total cell counts in these two conditions???

Minors:

-Legend of Figure 5: Please double check the order of the panels.

-In the result section it is cited Figure S5 and Supplemental Figure 6D, however these figures do not exist.

-In methods section, subheading "Surgical Procedure": It is described the cecal abration as one of the lesions to induce adhesion formation, but this procedure is not employed in this study.

Reviewer #3:

Remarks to the Author:

The submission by Zindel et al examined the effect of microbial contamination within the peritoneum on post-surgical adhesion formation. The authors used a model where a peritoneal button was created, followed by CLP-induced sepsis. The authors show the PB/CLP combination led to the greatest increase in adhesion formation. There was also a prominent role played by EGFR in this process, as inhibition of this signaling pathway reduced adhesion formation. Importantly, the authors also showed many of the same features were seen in humans with appendicitis.

Overall, this is a nice study. It is well written, and the conclusions reached are supported by the data presented. There are a few items the authors should consider, which could elevate the manuscript further.

1. In Figure 1, the authors compare germ-free mice to conventional SPF mice. It is well-known that GF mice have defects in immune system development and composition. What role does the immune system play in the adhesion formation? CLP leads to an influx of immune cells into the peritoneum. What happens when only the PB surgery is performed? Does the immune cell influx become altered during PB/CLP surgery?

2. The authors conclusion of Aim 1 is that "contamination of live gut microbes, rather than intestinal content, into the peritoneal compartment enhances the formation of post-surgical adhesions." The authors should consider using a cecal slurry model with live vs. heat-killed gut microbes to provide direct evidence to support this conclusion. In addition, what would happen during true sterile inflammation - such as LPS injection into the peritoneum after PB formation?

3. Sepsis patients are typically treated with antibiotics. Can adhesion formation be reduced when systemic antibiotics are administered after CLP?

Reviewer #4:

Remarks to the Author:

Authors established an animal model for evaluating the peritoneal adhesion caused by surgical injury in the presence or absence of bacterial contamination. They demonstrated the major contribution of bacterial contamination on the adhesion and collagen deposition. They further found that EGFR ligands produced mainly by leukocytes activated mesothelial cells and caused MMT for myofibroblasts which secrete collagen. They showed treatment with gefitinib, an EGFR inhibitor, prevented the adhesion. Finally, they showed that their mechanisms could be involved in human patients who experienced adhesion after surgery. This study was well designed and performed carefully. However, methods used were not stated appropriately. This reviewer has several concerns as follows.

Specific comments

1. Several data were present or explained inappropriately. There were discordance between Figures 5 c, d, e and their legends. Figure S5B stated in line 238 was missing in the supplementary Materials.
2. Line 233. Authors stated "measured in the peritoneal lavage". It is unclear whether authors measured peritoneal lavage fluid or cells in the peritoneal lavage fluid. Authors demonstrated the elevated expression of EGFR-ligands by a multiplexed mesoscale cytokine/chemokine screening (Lines 231-232). This assay may measure cytokines at protein levels. However, the names of cytokines measured were stated by italic characters (line 244), these should mean genes, not proteins. Authors should make much clearer these statements.
3. Authors demonstrated that leukocytes were the major source of EGFR-ligands. Neutrophils accumulated into the peritoneal cavity after CLP. These results suggest that neutrophils may be the major source of EGFR-ligands. Comparison of neutrophils and monocyte/macrophages in the peritoneal cavity on EGFR-ligand production would strengthen the results of this study.
4. The methods for gefitinib administration were unclear. What diluent was used to solve 100mg/kg gefitinib? How much volume of the diluent was used? The concentration of gefitinib administered may be important, because gefitinib was injected into the peritoneal cavity. When the gefitinib treatment started? Immediately after surgery or 2~3 hours after the surgery?
5. The dose of gefitinib dose is important to translate the results of this study to clinic. The minimal gefitinib dose to inhibit adhesion should be determined.

Minor comments

1. Abstract should be shortened according to the instruction.
2. In the Discussion section, the spells of "EGFR" and "Egfr" were mixed. These should be unified appropriately.

Point by point answers to the Reviewers

Reviewer #1 (Remarks to the Author):

The authors show in their study that in abdominal injuries with bacterial contamination, mesothelial cells express increased levels of EGFR. The EGFR ligands are supplied by immune cells. Inhibition by an EGFR inhibitor reduces the murine adhesions but does not prevent them. The findings are further supported by patient data.

While the authors provide a solid foundation and evidence for a potential key role of EGFR dependent activation of mesothelial cells, and a potential immune-mesothelial crosstalk, the current version of the study lacks novelty.

EGFR possesses a multitude of downstream effectors and mediators, which affect a variety of cells (e.g. also fibroblasts). But what is the effect on mesothelial cells which leads to adhesion development is understudied here. Furthermore, the authors do not show the **activation state of the EGFRs found in mesothelial cells (e.g. phosphorylation Tyr1045)**. For example, EGFRs also regulate vesicular transport, which in turn plays a key role in the biology of mesothelial cells. The following comments would add value to a revised study:

Major comments:

- Genetic tracing has been shown, even in combination with genetic depletion strategies such as DTA (Fischer et al., 2020), and could be mentioned.

Thank you for pointing this out. We now mention this. See below in bold:

The capacity of mesothelial cells to undergo a mesothelial to mesenchymal transition has been reported in other diseases¹⁻⁴. Furthermore, it has been suggested that mesothelial cells are an important cellular origin of adhesions, as shown with membrane dyes⁵⁻⁷. **More recently, Fischer et al. used an inducible genetic lineage tracing system based on the mesothelial cell marker *Procr* to show that adhesion myofibroblasts arose from mesothelial precursors⁸. Here, we used a *Wt1* based genetic lineage tracing system which aligns with these finding and demonstrate that the vast majority of myofibroblasts within adhesions are derived from the mesothelium, and not from fibroblasts. Therefore, inhibiting the molecular mechanisms by which mesothelial cells become activated myofibroblasts may provide a means to ameliorate the major source of collagen found in adhesions.**

- The transcriptional changes need to be validated at the protein level. It is essential to search for active signaling molecules, as claimed by the authors. Phosphorylated EGFR should be visualized together with mesothelial markers. The cells in the adhesion sites should show increased phospho-EGFR signalling compared to the mesothelial cells in the neighboring sites. This is of essential importance, since EGFR plays a key role in many processes.

Thank you for this input. We performed a series of immunohistochemistry experiments to demonstrate that upregulation of pEGFR is specific for the injured areas. Furthermore, we found that pEGFR signal co-localizes well with mesothelial markers such as Podoplanin and GMP6A. We amended the text and figures as follows (changes in bold):

Next, we questioned what was driving proliferation of the mesothelium. Examination of the gene ontology network node “Proliferation/Activation” pointed to receptor tyrosine kinases (RTK) signaling as potential core driver for the observed changes in mesothelial cells (Figure 3D). Among all RTKs, *ErbB2* showed the highest differential expression (log2 fold change = 2.1, $p < 0.001$) when comparing germ-free with SPF mice. Similarly, EGFR (*ErbB1*), was significantly increased after surgery in SPF mice when compared with germ-free mice (Fig 4C). In addition, the respective downstream pathways of EGFR and *ErbB2* were highly differentially expressed (Fig. 4a,b), including the MAPK pathway (Fig. 4b). **We next sought to confirm the upregulation and activation of EGFR in mesothelial cells in response to injury and bacterial contamination at the protein level. Interestingly, 24 hours after injury, very few mesothelial cells remained within the peritoneal button injury (Fig. 4d). Therefore, we hypothesized that the increased mesothelial *Egfr* expression (Fig. 4a) must come from mesothelial cells that were isolated**

from regions adjacent to the injury. Indeed, the mesothelium within a few millimeters of peritoneal injuries showed a large increase of EGFR signaling as indicated by the activated form of EGFR (pEGFR) (Fig. 4e) whereas no pEGFR expression was found in distant mesothelium. Importantly, pEGFR showed a high degree of co-localization with the mesothelial cell marker Podoplanin, suggesting that the observed increase of pEGFR is specific for mesothelial cells. Furthermore, EGFR expression was still elevated seven days after injury in SPF but not GF mice (Fig. S4b). Taken together, both RNA and protein data suggest that EGFR-signaling is specifically activated in mesothelial cells. This activation occurs in response to injury and is potentiated by microbial contamination.

Interestingly, ERK signaling does not inhibit adhesion formation in vivo (Fischer et al., 2020 supplementary data). Therefore, the EGFR-dependent downstream effects should be described in more detail and the mode-of-action of EGFR-mediated adhesion formation should be described.

We went back and investigated the downstream effects of Amphiregulin and HB-EGF in mesothelial cells. In vitro, we found an activation of the ERK/MAPK pathway and the PI3K/Akt pathway but not the Stat3 pathway. See below in bold:

Next, we found that both AREG and HB-EGF led to a significant and dose-dependent increase of EGFR Phosphorylation (pEGFR) in cultured primary mesothelial cells (Fig 5g). **We went on to interrogate what downstream pathways played a role in our model in comparison with the sterile model by Fischer et al, where Erk did not play a role⁸. In our system, the increase of pEGFR in turn activated MAPK/Erk pathway (Fig. 5g) and higher ligand concentrations led to an activation of the PI3K/Akt pathway (Fig. 5g).** We observed no

activation of the Stat3 pathway even with high ligand doses (Fig 5g).

We then went on and investigated the ERK/MAPK pathway and PI3K/Akt pathway *in vivo* by using selective PI3K and Mek inhibitors respectively. This confirmed that in our model Erk signaling played an important role in adhesion formation. Since this resulted in some significant changes in Fig. 6 we have added the whole figure and the corresponding changes in the text (relevant changes in bold) below:

Next, asked whether pharmacological inhibition of EGFR can be exploited to prevent adhesion formation. **Gefitinib was used to inhibit the phosphorylation of EGFR (Fig. 6a, b) and Selumetinib and Ly294002 were used to inhibit the downstream kinases Mek and PI3K respectively (Fig. 6a, b). In vitro, Gefitinib led to a significant reduction of the collagen production (Fig. 6c) and migration (Fig. 6d-f) of mesothelial cells.** In vivo, the daily intraperitoneal administration of 100 mg/kg of Gefitinib⁹ resulted in a significant reduction of post-surgical adhesion formation (Fig. 6g). In addition, intraperitoneal treatment with Gefitinib resulted in a significant reduction of tdTomato positive mesothelium derived cells within adhesions (Fig. 6h, i)). We also investigated administration of Gefitinib by oral gavage with either 20 or 100 mg/kg daily or a once weekly dose of 400mg/kg as previously described⁹. We found that oral application of Gefitinib or intraperitoneal doses of less than 50mg/kg per day did not significantly reduce the adhesion index (Fig. S8a, b). This would suggest the need for a high local concentration (μ M range) to be effective. **Using other kinase inhibitors such as the Mek inhibitor Selumetinib and PI3K inhibitor Ly294002 showed that inhibition of the MAPK/Erk but not the PI3K/Akt pathway results in a reduction of postoperative adhesions.** Taken together, these findings suggest that EGFR signals through the MAPK/Erk pathway potentiate post-surgical adhesion formation.

- The authors claim that EGFR-dependent activation of mesothelial cells does not occur through EGF but through ligands such as AREG, EREG or TNFalpha. Which exact ligands are involved and who is secreting these ligands in mouse and human adhesions. SCRNAseq data and immunostaining could eliminate these points.

We performed scRNA-Seq experiments of peritoneal button biopsies at 24h post-surgery to adress the reviewer. This allowed us to identify the cells producing EGFR ligands and also the ligands that were produced within the wounds themselves. In light of these new results we have significantly amended Figure 5 and the corresponding text (relevant changes in bold letters):

However, through multiple assays we were unable to detect EGFR ligands in the peritoneal fluid. We have recently shown that macrophages can be recruited to peritoneal injuries by a direct route from the peritoneal cavity¹⁰. Furthermore, a series of recent reports highlights the emerging role immune cell EGFR production in the regulation of inflammation and tissue repair¹¹⁻¹³. To explore whether EGFR ligands were produced in a paracrine fashion by immune cells that infiltrate the peritoneal injury, we dissociated peritoneal injury biopsies into single cell suspensions and performed single cell RNA-Sequencing (Fig. 5a). Manually annotated (Seurat) and automatically annotated (SingleR) clustering confirmed the presence of mesothelial cells (*Krt19*⁺, *Gpm6A*⁺) and several distinct populations of CD45⁺ immune cells (Figs. 5b, S6a-c). As expected, the number of mesothelial cells within peritoneal buttons was very small in comparison to the number of infiltrating

immune cells (Fig. 4d and Fig. 5c, d). This analysis showed that mesothelial cells were the only cells that expressed *Egfr* (Fig. 5c) but did not express significant amounts of EGFR ligands (Fig. S7b). Within injuries, the main ligands with known activity on EGFR homo- and hetero-dimers were Heparin-binding Epidermal Growth Factor (HB-EGF) and AREG (Fig. 5d, S7b). The major HB-EGF and AREG producers were bone marrow derived macrophages (Figure 5d). Interestingly, a small subset of B-cells also expressed AREG. This B-cell subset, characterized by the expression of *Ly6d*, *Cd79a*, *Ms4a1* and *E330020D12Rik*, was only present in mice that underwent CLP in addition to injury (PB).

Fig 5

-A cell-type specific depletion or cell-type deletion of the ligands should be performed.

We recognize that a cell-type specific depletion or deletion of the ligands would provide the ultimate experimental evidence in support of the proposed mechanism. However, considering our qPCR and scRNA-Seq data, we must assume that more than one cell type provides mesothelial cells with EGFR ligands. Potential players include Neutrophils, monocyte derived macrophages and B-cells.

In our hands, Neutrophil depletion (1A8 antibody) and B/T cell depletion (Rag2^{-/-}) does not result in a change of adhesion score. It is important to note that neutrophil depletion in other models/labs has been shown to decrease adhesion formation¹⁴. Furthermore, monocyte deficient CCR2 KO mice show an increase of adhesions (it also results in an increased neutrophil influx). Tsai and colleagues¹⁴ recently showed that the enrichment of monocytes results in a decrease of adhesion formation which agrees with these CCR2-KO data. In summary, the mechanisms underlying inflammatory cell kinetics, as Tsai et al have recently shown, play critical roles in mouse peritoneal adhesion formation. But they are very complex and only partially resolved, and as such are beyond the scope of this study.

- Is it possible that these ligands activate a distinct EGFR signaling pathway induced by EGF in mesothelial cells?

Indeed, we went back and showed that these ligands activate the MAPK/Erk pathway and to a smaller degree the PI3K/Akt pathway (see also our answer to question above). As requested, we also tested EGF (in addition to AREG and HBEGF). The effect of EGF is comparable, albeit at lower doses, to that of HB-EGF and Areg (data added to Fig. S5).

- Bacterial contamination takes place throughout the entire abdomen, why do adhesions still only occur locally? this could be better explained. Adhesion induced by EGFR alone could therefore be a partial mechanistic story. The authors should therefore investigate how different injury +/- EGFR affects mesothelial cells.

Thank you for pointing this out. Indeed, it is very interesting that the contamination takes place everywhere yet the effect on adhesions is only in areas that were also injured. We have added some experiments using Cecal Slurry (CS) instead of CLP, resulting in an even more homogeneous contamination of the whole peritoneal cavity. Yet, not

once have we seen adhesions forming in non-injured areas. To make this point clearer we have added the following text (relevant changes in bold):

Next, we replaced the CLP with cecal slurry (CS) that was generated from feces of SPF mice. The effect of both, native and heat inactivated CS on adhesion formation was comparable to CLP (Fig. S1c). **Interestingly, the microbial contamination (CLP, CS) takes place throughout the entire peritoneum, yet adhesions only occurred locally at the site of injury.** The administration of Lipopolysaccharide (LPS) also resulted in an increase of adhesions over injury only, but the effect was smaller than that of CLP or CS (Fig. S1c). Taken together, these data suggest that contamination with gut microbes, rather than intestinal content, drives the formation of post-surgical adhesions.

- The treatment regimen with gefitinib raises several questions. 100mg/kg means a daily injection of 2mg/20g mouse. These amounts are unrealistically high for clinical use and could be due to indirect effects, other than on mesothelium. It would therefore be desirable if the authors could show another EGFR inhibitor. Also a titration curve needs to be shown what are the minimal doses that are effective.

We would like to partially disagree on this point. For a short-term use a dose of 100mg/kg is not completely unrealistic for short-term clinical use. Gefitinib (Iressar®) has been approved for routine clinical use in non-small cell lung cancer (NSCLC). It is true that the standard dose for long-term treatment with Iressa is 250mg per day (3.5mg/kg/day assuming a 70kg standard patient) which is one order of magnitude lower than the dose we used to prevent adhesions. However, it is well documented that in rodents higher doses (up to 80mg/kg/day) are safe (short term) and have an additional benefit against tumors⁹.

We went back and performed some additional experiments. Using in vitro wound healing and collagen deposition assays with primary mesothelial cells, we determined that the minimal effective dose of Gefitinib was in the micromolar range (**Data added to Figure 5**).

We agree that this is a rather high local concentration which also explains why the intraperitoneal dose worked when the oral dose did not. We then performed a dose titration in vivo to further confirm that an intraperitoneal dose of at least 50mg/kg Gefitinib was necessary to reduce adhesion formation (**Data added to Fig S8**):

Fig S8

In conclusion, the proposed dose is not completely unrealistic, yet it is a limitation of this approach. Hence, we propose to add the following text (change in bold) to the discussion:

This is particularly interesting because several small molecule EGFR-inhibitors, like Gefitinib used in this study, have already been approved for treatment of non-small lung cancer. **In our study, high Gefitinib concentrations were necessary to be effective on mesothelial cells in vitro and in vivo. This is a potential limitation to this approach.** Further studies, such as retrospective analysis of patients that underwent abdominal surgery while being treated with EGFR-inhibitors as well as prospective studies are warranted to investigate the benefit of EGFR-inhibition in preventing adhesions in human patients.

As the reviewer suggested we went ahead and investigated additional inhibitors downstream of EGFR. See below in bold the corresponding changes to manuscript and the new panel in Fig. 6.

Using other kinase inhibitors such as the Mek inhibitor Selumetinib and PI3K inhibitor Ly294002 showed that inhibition of the MAPK/Erk but not the PI3K/Akt pathway results in a reduction of postoperative adhesions (Fig. 6g).

-A much deeper analysis of the pathway and mode of action needs to occur for this to be novel.

We have now included data from a successful scRNA-Seq experiment and provide an in-depth characterization of the intracellular pathways in primary mesothelial cells that were treated with EGFR-agonists and different inhibitors. The role of EGFR signaling in mesothelial to mesenchymal transition of mesothelial cells is novel as are the therapeutic complications this might have in post-surgical adhesion formation.

-Can ectopic ligand administration or EGFR activation induce adhesion formation without injury?

We went back to the lab and investigated whether we could replicate the effect of bacterial contamination (CLP) by combining PB with topical administration of EGFR ligands. We used both recombinant mouse Amphiregulin and recombinant EGF. However, the application of EGFR ligands alone was not enough to induce adhesion formation without injury. The following text (changes in bold) and figure was added:

Next, we tested whether EGFR agonists were sufficient to recapitulate the effect of bacterial contamination in our adhesion model. However, neither the injection of recombinant AREG nor recombinant EGF—which showed the strongest effect on mesothelial cells in vitro—were sufficient to increase the adhesion score in mice that underwent injury model (PB)(Fig. S5h).

-No direct proof of which immune cells express and secrete ligands are responsible potentially for a immune-mesothelial crosstalk. Depletion schemes or purification and transplantation of immune cells needs to be done.

As we have shown and discussed in response to the previous comment the effect of peritoneal leukocyte kinetics on adhesion formation is rather complex. The major producers of AREG and HB-EGFR are bone marrow derived macrophages. All macrophages (including the resident Gata6+ population) can be depleted using clodronate liposomes. Combining clodronate liposomes with our model (PB + CLP) results in a significant reduction of adhesions. However, in light of our recent paper¹⁰ and that of Tsai et al¹⁴ this effect of clodronate is probably not only due to the loss of EGFR ligands. As we stated above, the role of leukocyte dynamics in adhesion formation is very complex and only partially resolved, and as such are beyond the scope of this study. Given the fact that several cell populations produce EGFR-ligands and that several ligands are at play, breeding a cell-type specific knock out is unlikely to work.

Minor comments:

- Supplemental figure 6; wrong title

Amended

Reviewer #2 (Remarks to the Author):

General comment:

This manuscript deals with an important medical complication, the peritoneal adhesions, which occur very frequently after abdominal surgery. Nowadays, there are no definitive treatments to prevent or to ameliorate the formation of adhesions. In addition, the cellular nature and the molecular mechanisms for adhesion formation are still open questions. In previous publications it has been demonstrated that targeting mesothelial to mesenchymal transition reduced adhesion formation in a sterile injury model. Nevertheless, abdominal surgery is not a sterile trauma and is often complicated by the contamination with gut microbes.

In this study, the authors hypothesized that microbe-induced inflammation contributes to peritoneal adhesion formation. The group employed various mouse models of peritoneal adhesion based on ischemic button, accompanied or not by cecal ligation and puncture and they demonstrate that the release of luminal contents, microbes, into the peritoneal cavity exacerbates the formation of peritoneal adhesions.

By using different experimental approaches, including lineage tracing, immunohistochemical studies and RNA-seq of purified mesothelial cells, the authors demonstrate that mesothelial cells are the main source of myofibroblasts via mesothelial to mesenchymal transition (MMT). Then the authors attempt to connect the MMT process with EGFR-signaling, which is upregulated in the presence of gut microbes. In this context, Gefitinib, a small molecule inhibitor of the EGFR-signaling reduced peritoneal adhesions in the mouse model. Finally, these findings have been partially recapitulated in biopsies from human patients.

Although the results presented are potentially interesting, the experimental approaches are technically sound, and data support the authors conclusions, the study lacks enough mechanistic analysis of the role of EGFR-signaling in both MMT and microbe-induced adhesions.

Specific comments:

1.- No mechanistic studies were performed to define how EGFR ligands induce the mesenchymal conversion of the mesothelial cells. In vitro studies with primary mesothelial cells to analyse the EGFR-mediated MMT and the mechanism of action of Gefitinib would strength author's conclusions.

Thank you for pointing this out. We have added a whole series of experiments addressing this question as requested by the reviewer. We would like to start by pointing out that the mere process of isolating primary mesothelial cells and plating them in vitro results in a partial activation with loss of epithelial and increase of mesenchymal markers. Since in vitro cultivation is a strong inducer of mesenchymal-ness, we were unable demonstrate a further increase of mesenchymal markers in vitro by adding EGFR ligands. However, we were able to show a loss of epithelial markers when giving EGFR ligands. Furthermore, we were able to inhibit EGFR driven MMT by administration of Gefitinib, further confirming the role of EGFR in MMT. We have amended the manuscript accordingly (**relevant changes in bold**):

Next, asked whether pharmacological inhibition of EGFR can be exploited to prevent adhesion formation. Gefitinib was used to inhibit the phosphorylation of EGFR (Fig. 6a, b) and Selumetinib and Ly294002 were used to inhibit the downstream kinases Mek and PI3K respectively (Fig. 6a, b). In vitro, Gefitinib led to a significant reduction of the collagen production (Fig. 6c) and migration (Fig. 6d-f) of mesothelial cells. **Furthermore, Gefitinib was able to inhibit EGFR-induced MMT in cultured primary mesothelial cells (Fig S8a)**. In vivo, the daily intraperitoneal administration of 100 mg/kg of Gefitinib⁹ resulted in a significant reduction of post-surgical adhesion formation (Fig. 6g). In addition, intraperitoneal treatment with Gefitinib resulted in a significant reduction of tdTomato positive mesothelium derived cells within adhesions (Fig. 6h, i)

Fig S8

2.-In the result section, there is a subheading entitled: “Gefitinib, a small molecule inhibitor of EGFR reduces wound healing of mesothelial cells in vitro and post-surgical adhesion formation in vivo”. However, these in vitro experiments are not shown.

We apologize for this mistake that has now been amended to:

Gefitinib, a small molecule inhibitor of EGFR reduces collagen deposition and MMT in vitro and post-surgical adhesion formation in vivo

3.- The authors present a detailed RNA seq and qPCR studies of mesothelial cells isolated by magnetic beads. They have found that a large number of genes are deregulated in the adhesion-inducing conditions at different time points. In figure 4, the authors demonstrate, at the level of gene expression, the activation of the EGFR pathway, at different times, after surgical damage. Likewise, they show that contamination with microbes induces a hyper-regulation of EGF receptors. However, it is well known that signal transmission through kinase pathways is regulated by post-translational processes including phosphorylation and subcellular localization of kinases. Therefore, a characterization of protein expression and/or post-translational modifications would significantly improve this study.

Thank you for pointing this out. We have added a series of in vitro studies using primary mesothelial cells stimulated with AREG and HB-EGF, the two major ligands we have identified in our system. In consequence, the following text and figures were added (changes in bold letters):

Next, we showed that both AREG and HB-EGF led to a significant and dose-dependent increase of EGFR Phosphorylation (pEGFR) in cultured primary mesothelial cells (Fig 5g). Increase of pEGFR in turn activated MAPK/Erk pathway (Fig. 5g) and in higher doses to an activation of the PI3K/Akt pathway (Fig. 5g). We observed no activation of the Stat3 pathway even with high ligand doses (Fig 5g). Epidermal growth factor (EGF), which was not expressed in our system in our scRNA-Seq experiment, resulted in an even stronger effect in vitro when compared with HB-EGF (Fig. S5g).

4.- Surprisingly, they found the up regulation of E-cadherin at early time points post-surgery, in parallel with the up-regulation of the transcriptional repressor Snail (Suppl. Figure 4). In addition, cultured mesothelial cells isolated from peritoneal adhesions showed a round morphology (Suppl. Figure 2). These data are in sharp contrast with what is known in about the MMT process. These apparent discrepancies should be discussed.

One last experiment ongoing looking into the nuclear vs cytosolic localization of E-cadherin. It has been shown that nuclear translocation is associated with MMT in some cells (not necessarily downregulation). If this is not the case I will simply add a sentence in the discussion indicating that E-cadherin is not consistent with the remainder of MMT signature.

5.- Figure 5: A better characterization of the immune/inflammatory response to PB and PB+CLP would reinforce the conclusions. For example, what is the main Th (Th1, Th2, Th17) response in the different conditions? The changes of the inflammatory responses between PB and PB+CLP are only qualitative or both qualitative and quantitative?. What are the total cell counts in these two conditions???

The overall cell number in the peritoneal cavity is not significantly different between PB and PB+CLP. As such, the changes of the inflammatory response between PB and PB + CLP that we describe in Figure 5 are both qualitative and quantitative. We have added the total cell counts in Figure 5:

Minors:

- Legend of Figure 5: Please double check the order of the panels.
- In the result section it is cited Figure S5 and Supplemental Figure 6D, however these figures do not exist.
- In methods section, subheading “Surgical Procedure”: It is described the cecal ablation as one of the lesions to induce adhesion formation, but this procedure is not employed in this study.

Thank you for pointing out these mistakes. The Legend of the significantly revised Figure 5 has been double checked. We double checked that all figures were cited correctly in the manuscript. We removed the sentence about cecal ablation, a model.

Reviewer #3 (Remarks to the Author):

The submission by Zindel et al examined the effect of microbial contamination within the peritoneum on post-surgical adhesion formation. The authors used a model where a peritoneal button was created, followed by CLP-induced sepsis. The authors show the PB/CLP combination led to the greatest increase in adhesion formation. There was also a prominent role played by EGFR in this process, as inhibition of this signaling pathway reduced adhesion formation. Importantly, the authors also showed many of the same features were seen in humans with appendicitis.

Overall, this is a nice study. It is well written, and the conclusions reached are supported by the data presented. There are a few items the authors should consider, which could elevate the manuscript further.

1. In Figure 1, the authors compare germ-free mice to conventional SPF mice. It is well-known that GF mice have defects in immune system development and composition. What role does the immune system play in the adhesion formation? CLP leads to an influx of immune cells into the peritoneum. What happens when only the PB surgery is performed? Does the immune cell influx become altered during PB/CLP surgery?

We recognize that GF mice have some profound defects in immune system development and composition. We and others have shown that peritoneal immune cells play an important function in adhesion formation^{10,14}. We went back and added a characterization of inflammatory cells and cytokines of GF vs colonized mice. We propose to add the following text and supplementary figures (new text in bold).

Corresponding to a decrease in adhesion formation, GF animals showed a significantly less pronounced increase of pro-inflammatory cytokines after surgery when compared with sDMDm2 mice (Fig. S1c). Nonetheless, GF animals were able to mount an inflammatory response post-surgery, indicated by a profound influx of inflammatory leukocytes (neutrophils and monocytes) into the peritoneal cavity (Fig. S1d). However, when compared with colonized animals, the infiltration of leukocytes in GF animals consisted of more monocytes and less neutrophils (Fig. S1d).

As for the second part of this question of what happens when only PB surgery is performed: we recognize that this was poorly explained in the previous version of the manuscript. To highlight the comparison of PB and PB+CLP with regards to inflammatory mediators and leukocyte influx into the peritoneal cavity, we have amended this part in the manuscript (changes in bold):

We therefore hypothesized that EGFR ligands must be produced in a paracrine fashion by another cell type such as peritoneal leukocytes. **To investigate the difference of the inflammatory response between mice that underwent PB and PB + CLP we characterized** the post-surgical chemotactic signature in the peritoneal cavity using a multiplexed mesoscale cytokine/chemokine screening. Hierarchical clustering of the cytokine/chemokine signature measured in the peritoneal lavage uncovered a distinct proinflammatory neutrophil-recruiting cytokine signature in colonized mice that underwent CLP (Fig. S5c). This proinflammatory signature was well separated from GF mice

undergoing CLP and colonized mice receiving only PB without CLP (Fig. S5c). Interestingly, hierarchical clustering revealed that the cytokine signature observed in GF mice with PB + CLP closely resembled that of colonized mice receiving only the PB without CLP (Fig. S5c). **Next, we performed flow cytometric characterization of the leukocyte influx into the peritoneal cavity. Corresponding to the chemokine profile, PB + CLP led to a significantly increased neutrophil recruitment when compared with PB alone (Fig. S5d,e). On the other hand, sterile damage (PB) alone led to an increased influx of monocytes (Fig. S5d,e) whereas the number of macrophages and B-cells was similar in both conditions. The difference between PB and PB + CLP was similar to the difference observed between GF and sDMDMm2 observed earlier (Fig S1c,d).**

2. The authors conclusion of Aim 1 is that "contamination of live gut microbes, rather than intestinal content, into the peritoneal compartment enhances the formation of post-surgical adhesions." The authors should consider using a cecal slurry model with live vs. heat-killed gut microbes to provide direct evidence to support this conclusion. In addition, what would happen during true sterile inflammation - such as LPS injection into the peritoneum after PB formation?

We went back to the lab and addressed these points with additional experiments. The resulting data fit well and make the overall story better. We have added the following sentences in the manuscript (changes in bold) and suggest adding the following figure to the Supplemental Figure S1.

There was no significant difference between the two colonized groups, sDMDMm2 and SPF (Figure 1E). In addition, adhesions sampled from GF animals showed a decreased collagen content when compared with adhesions sampled from colonized mice (Figure 1F). **Next, we replaced the CLP with cecal slurry (CS) that was generated from feces of SPF mice. The effect of both, native and heat inactivated CS on adhesion formation was comparable to CLP (Fig. S1c). The administration of Lipopolysaccharide (LPS) also resulted in an increase of adhesions over injury only, but the effect was smaller than that of CLP or fecal slurry (Fig. S1c). Taken together,** these data suggest that contamination with **live** gut microbes—**dead or alive**--rather than intestinal content, drives the formation of post-surgical adhesions. **A process that is only partially driven by LPS.**

In the Methods section we added this:

Cecal slurry (CS) stock preparation

CS was prepared as previously described¹⁵. In brief, fecal content from ceca of C57/Bl6 mice was collected and mixed with sterile water at a ratio of 0.5ml water to 100mg of cecal content. The suspension was then filtered consecutively through a 100µm and 70µm filter. The filtered solution was then mixed with an equal volume of 30% glycerol in PBS, resulting in a final CS stock solution in 15% glycerol in PBS. The CS stock was aliquoted and stored at -80°C for later experiments. Heat inactivation of CS was performed by incubating CS stock solution for 20min at 72°C. Colony formation assays were performed before and after to confirm heat inactivation.

e

3. Sepsis patients are typically treated with antibiotics. Can adhesion formation be reduced when systemic antibiotics are administered after CLP?

We thank the reviewer for highlighting antibiotics as a potential therapeutic. This would certainly be a remarkably interesting avenue to improve this clinical problem. To investigate this possibility, we combined systemic administration of different broad-spectrum antibiotics with our model. Unfortunately, none of the tested antibiotics resulted in a significant reduction of adhesion formation. We have added this to manuscript and figures (relevant changes in bold):

Correspondingly, when mice were treated with broad-spectrum antibiotics prior to surgery, no reduction of the adhesion index was observed (Fig. S1f). This does not contradict the GF data were mice had no bacteria prior to surgery. Moreover, none of these regimens completely eradicate bacteria so a cocktail may be needed. Interestingly, the microbial contamination (CLP, CS) takes place throughout the entire peritoneum, yet adhesions only occurred locally at the site of injury. The administration of Lipopolysaccharide (LPS) also resulted in an increase of adhesions over injury only, but the effect was smaller than that of CLP or CS (Fig. S1e). Taken together, these data suggest that contamination with gut microbes rather than intestinal content, drives the formation of post-surgical adhesions.

f
In the methods section we have added:

No standard antibiotics prophylaxis was administered. If perioperative antibiotics were given, they were administered 30-60 min prior to surgery by subcutaneous route. The antibiotic substance given were either Ceftriaxone (120mg/kg), Clindamycine (36mg/kg) or Amoxicillin + Clavulanic acid (200 + 20 mg/kg).

Reviewer #4 (Remarks to the Author):

Authors established an animal model for evaluating the peritoneal adhesion caused by surgical injury in the presence or absence of bacterial contamination. They demonstrated the major contribution of bacterial contamination on the adhesion and collagen deposition. They further found that EGFR ligands produced mainly by leukocytes activated mesothelial cells and caused MMT for myofibroblasts which secrete collagen. They showed treatment with gefitinib, an EGFR inhibitor, prevented the adhesion. Finally, they showed that their mechanisms could be involved in human patients who experienced adhesion after surgery. This study was well designed and performed carefully. However, methods used were not stated appropriately. This reviewer has several concerns as follows.

Specific comments

1. Several data were present or explained inappropriately. There were discordance between Figures 5 c, d, e and their legends. Figure S5B stated in line 238 was missing in the supplementary Materials.

This has been amended.

2. Line 233. Authors stated “measured in the peritoneal lavage”. It is unclear whether authors measured peritoneal lavage fluid or cells in the peritoneal lavage fluid. Authors demonstrated the elevated expression of EGFR-ligands by a multiplexed mesoscale cytokine/chemokine screening (Lines 231-232). This assay may measure cytokines at protein levels. However, the names of cytokines measured were stated by italic characters (line 244), these should mean genes, not proteins. Authors should make much clearer these statements.

We realize that this paragraph has been unclear, and we have significantly amended this section accordingly (relevant changes in bold letters):

Hierarchical clustering of the cytokine/chemokine signature measured in the peritoneal lavage **fluid** uncovered a distinct proinflammatory neutrophil-recruiting cytokine signature in colonized mice that underwent CLP (Fig. S5c). This proinflammatory signature was well separated from GF mice undergoing CLP and colonized mice receiving only PB without CLP (Fig. S5c). Interestingly, hierarchical clustering revealed that the cytokine signature observed in GF mice with PB + CLP closely resembled that of colonized mice receiving only the PB without CLP (Fig. S5c). **Next, we performed flow cytometric characterization of the leukocyte influx into the peritoneal cavity.** Corresponding to the chemokine profile, PB + CLP led to a significantly increased neutrophil recruitment when compared with PB alone (Fig. S5d,e).

(...)

Next, we isolated peritoneal leukocytes 24 hours post-surgery and found that peritoneal leukocytes isolated from colonized mice after CLP showed a significantly increased expression (**quantitative PCR**) of the EGFR ligands *Areg*, *Ereg*, and *Tgfa* when compared with peritoneal leukocytes isolated from mice without intraperitoneal microbe challenge (Fig. S5f). **These findings suggested that contamination of the peritoneal compartment with live gut microbes leads to an increase in leukocyte recruitment, which produce EGFR ligands in the peritoneal cavity fluid. However, through assays we were unable to detect EGFR ligands in the peritoneal fluid.**

3. Authors demonstrated that leukocytes were the major source of EGFR-ligands. Neutrophils accumulated into the peritoneal cavity after CLP. These results suggest that neutrophils may be the major source of EGFR-ligands. Comparison of neutrophils and monocyte/macrophages in the peritoneal cavity on EGFR-ligand production would strengthen the results of this study.

We did as the reviewer asked and performed a series of experiments in which we sorted monocytes, macrophages and neutrophils that we isolated from the peritoneal cavity. We plated these cells and were not able to detect significant amounts of EGFR-ligands in the supernatant. Therefore, we performed single cell digestions of the peritoneal injuries (buttons) and performed scRNA-Seq to investigate in an untargeted manner what immune cell population was producing what EGFR ligand. We found that the major source of EGFR-ligands was in fact not the neutrophils but monocyte derived macrophages that infiltrated the wounds. This discovery led us to make some significant changes in the manuscript and figures. Please find these changes below (changes are highlighted in bold):

EGFR-ligands are produced by bone-marrow derived macrophages and a B-cell subset that are recruited to the wound.

We next asked the question what molecules ligate to EGFR and induce its activation post-surgery. EGFR expression seemed to be predominantly on the basolateral side of the healthy mesothelium (Fig. S5a). We initially hypothesized that surgical disruption of the mesothelial integrity may expose the basolateral receptor to the ligand available in the peritoneal cavity. This hypothesis was further supported by the observation that proliferative mesothelial cells were near the sites of surgical injury (Figs. 2k, S2f-h). Mesothelial cells produce a certain amount of EGFR ligands in an autocrine fashion (Fig. S5b). However, no difference in EGFR mesothelial ligand transcripts was detected over time and when comparing mice with (SPF) and without (GF) intraperitoneal microbe challenge three hours post-surgery. Except for transforming growth factor alpha (*TGF α*) which showed a decreased production (Fig. S5b). We therefore hypothesized that EGFR ligands must be produced in a paracrine fashion by another cell type such as peritoneal leukocytes. To investigate the difference of the inflammatory response between mice that underwent PB and PB + CLP we started characterizing the post-surgical chemotactic signature in the peritoneal cavity using a multiplexed mesoscale cytokine/chemokine screening. Hierarchical clustering of the cytokine/chemokine signature measured in the peritoneal lavage uncovered a distinct proinflammatory neutrophil-recruiting cytokine signature in colonized mice that underwent CLP (Fig. S5c). This proinflammatory signature was well separated from GF mice undergoing CLP and colonized mice receiving only PB without CLP (Fig. S5c). Interestingly, hierarchical clustering revealed that the cytokine signature observed in GF mice with PB + CLP closely resembled that of colonized mice receiving only the PB without CLP (Fig. S5c). Next, we performed flow cytometric characterization of the leukocyte influx into the peritoneal cavity. Corresponding to the chemokine profile, PB + CLP led to a significantly increased neutrophil recruitment when compared with PB alone (Fig. S5d,e). On the other hand, sterile damage (PB) alone led to an increased influx of monocytes (Fig. S5d,e) whereas the number of macrophages and B-cells was similar in both conditions. The difference between PB and PB + CLP was similar to the difference observed between GF and sDMDMm2 observed earlier (Fig S1c,d). Next, we isolated peritoneal leukocytes 24 hours post-surgery and found that peritoneal leukocytes isolated from colonized mice after CLP showed a significantly increased expression (quantitative PCR) of the EGFR ligands *Areg*, *Ereg*, and *Tgfa* when compared with peritoneal leukocytes isolated from mice without intraperitoneal microbe challenge (Fig. S5f). These findings suggested that contamination of the peritoneal compartment with live gut microbes leads to an increase in leukocyte recruitment, which produce EGFR ligands in the peritoneal cavity. However, by using different assays we were unable to detect EGFR ligand proteins in the peritoneal fluid. We have recently shown, that macrophages can be recruited to peritoneal injuries by a direct route from the peritoneal cavity¹⁰. Furthermore, a series of recent reports highlights the emerging role immune cell EGFR production in the regulation of inflammation and tissue repair¹¹⁻¹³. To explore whether EGFR ligands were produced in a paracrine fashion by immune cells that infiltrate the peritoneal injury, we dissociated peritoneal injury biopsies into single cell suspensions and performed single cell RNA-Sequencing (Fig. 5a). Manually annotated (Seurat) and automatically annotated (SingleR) clustering confirmed the presence of mesothelial cells (*Krt19*⁺, *Gpm6A*⁺) and several distinct populations of CD45⁺ immune cells (Figs. 5b, S6a-c). The number of mesothelial cells within peritoneal buttons was—as expected—very small in comparison to the number of infiltrating immune cells (Fig. 4d and Fig. 5c, d). The mesothelial cells were the only cells that expressed *Egfr* (Fig. 5c) but they did not express significant amounts of EGFR ligands (Fig. S7b). Within injuries, the main ligands with known activity on EGFR homo- and hetero-dimers were Heparin-binding Epidermal Growth Factor (HB-EGF) and AREG (Fig. 5d, S7b). The major HB-EGF and AREG producers were bone marrow derived macrophages. Interestingly, a small subset of B-cells also expressed AREG. This B-cell subset, characterized by the expression of *Ly6d*, *Cd79a*, *Ms4a1* and *E330020D12Rik*, was only present in mice that underwent CLP in addition to injury (PB). Next, we showed that both AREG and HB-EGF led to a significant and dose-dependent increase of EGFR Phosphorylation (pEGFR) in cultured primary mesothelial cells (Fig 5g). Increase of pEGFR in turn activated MAPK/Erk pathway (Fig. 5g) and in higher doses to an activation of the PI3K/Akt pathway (Fig. 5g). We observed no activation of the Stat3 pathway even with high ligand doses (Fig 5g). Epidermal growth factor (EGF), which was not expressed in our system in our scRNA-Seq experiment, resulted in an even stronger effect in vitro when compared with HB-EGF (Fig. S5g). Next, we tested whether EGFR agonists were able to recapitulate the effect of bacterial contamination in our adhesion model. However, neither the injection of AREG nor EGF was able to increase the adhesion score in mice that underwent injury model (PB).

Fig 5

4. The methods for gefitinib administration were unclear. What diluent was used to solve 100mg/kg gefitinib? How much volume of the diluent was used? The concentration of gefitinib administered may be important, because gefitinib was injected into the peritoneal cavity. When the gefitinib treatment started? Immediately after surgery or 2~3 hours after the surgery?

Thank you for pointing this out. We have amended the manuscript in the methods section.

Preparation and administration of small molecule inhibitors

Gefitinib and Ly294002: Stock solutions were prepared by dissolving 100mg/ml in DMSO. Stock solutions were diluted with Saline to reach a final concentration of 20mg/ml Gefitinib in 20% DMSO. Selumetinib was dissolved in 10% DMSO in corn oil. Small molecule inhibitors were administered 2-3 hours after the surgery and once daily thereafter. Gefitinib and Ly294002 were administered by intraperitoneal injection and Selumetinib by oral gavage, with the doses as specified in the manuscript and figures.

5. The dose of gefitinib dose is important to translate the results of this study to clinic. The minimal gefitinib dose to inhibit adhesion should be determined.

Thank you for pointing this out. We have done these experiments as requested and added the following changes to the manuscript and figures (changes in bold letters):

We also investigated administration of Gefitinib by oral gavage with either 20 or 100 mg/kg daily or a once weekly dose of 400mg/kg as previously described ⁹. We found that oral application of Gefitinib **or intraperitoneal doses of less than 50mg/kg per day did not significantly reduce the adhesion index (Fig. S8b, c)**. This would suggest the need for a high local concentration (µM range) to be effective.

Minor comments

1. Abstract should be shortened according to the instruction.
2. In the Discussion section, the spells of “EGFR” and “Egfr” were mixed. These should be unified appropriately.

The abstract has been revised and care was taken to unify the use of EGFR throughout the whole manuscript.

References

- 1 Chen, Y. T. *et al.* Lineage tracing reveals distinctive fates for mesothelial cells and submesothelial fibroblasts during peritoneal injury. *Journal of the American Society of Nephrology : JASN* **25**, 2847-2858, doi:10.1681/asn.2013101079 (2014).
- 2 Li, Y., Wang, J. & Asahina, K. Mesothelial cells give rise to hepatic stellate cells and myofibroblasts via mesothelial–mesenchymal transition in liver injury. *Proceedings of the National Academy of Sciences* **110**, 2324-2329, doi:10.1073/pnas.1214136110 (2013).
- 3 Mutsaers, S. E. *et al.* Mesothelial cells in tissue repair and fibrosis. *Frontiers in Pharmacology* **6**, 113, doi:10.3389/fphar.2015.00113 (2015).
- 4 Mutsaers, S. E., Prele, C. M., Pengelly, S. & Herrick, S. E. Mesothelial cells and peritoneal homeostasis. *Fertility and sterility* **106**, 1018-1024, doi:10.1016/j.fertnstert.2016.09.005 (2016).
- 5 Tsai, J. M. *et al.* Surgical adhesions in mice are derived from mesothelial cells and can be targeted by antibodies against mesothelial markers. *Science Translational Medicine* **10**, doi:10.1126/scitranslmed.aan6735 (2018).
- 6 Sandoval, P. *et al.* Mesothelial-to-mesenchymal transition in the pathogenesis of post-surgical peritoneal adhesions. *The Journal of pathology* **239**, 48-59, doi:10.1002/path.4695 (2016).
- 7 Rinkevich, Y. *et al.* Identification and prospective isolation of a mesothelial precursor lineage giving rise to smooth muscle cells and fibroblasts for mammalian internal organs, and their vasculature. *Nature cell biology* **14**, 1251-1260, doi:10.1038/ncb2610 (2012).
- 8 Fischer, A. *et al.* Post-surgical adhesions are triggered by calcium-dependent membrane bridges between mesothelial surfaces. *Nature communications* **11**, 3068, doi:10.1038/s41467-020-16893-3 (2020).
- 9 Zhang, Q. *et al.* Effect of weekly or daily dosing regimen of Gefitinib in mouse models of lung cancer. *Oncotarget* **8**, 72447-72456, doi:10.18632/oncotarget.19785 (2017).
- 10 Zindel, J. *et al.* Primordial GATA6 macrophages function as extravascular platelets in sterile injury. *Science (New York, N.Y.)* **371**, eabe0595, doi:10.1126/science.abe0595 (2021).
- 11 Ko, J. H., Kim, H. J., Jeong, H. J., Lee, H. J. & Oh, J. Y. Mesenchymal Stem and Stromal Cells Harness Macrophage-Derived Amphiregulin to Maintain Tissue Homeostasis. *Cell reports* **30**, 3806-3820.e3806, doi:10.1016/j.celrep.2020.02.062 (2020).
- 12 Minutti, C. M. *et al.* A Macrophage-Pericyte Axis Directs Tissue Restoration via Amphiregulin-Induced Transforming Growth Factor Beta Activation. *Immunity* **50**, 645-654.e646, doi:10.1016/j.immuni.2019.01.008 (2019).
- 13 Zaiss, D. M. W., Gause, W. C., Osborne, L. C. & Artis, D. Emerging functions of amphiregulin in orchestrating immunity, inflammation, and tissue repair. *Immunity* **42**, 216-226, doi:10.1016/j.immuni.2015.01.020 (2015).
- 14 Tsai, J. M. *et al.* Neutrophil and monocyte kinetics play critical roles in mouse peritoneal adhesion formation. *Blood Advances* **3**, 2713-2721, doi:10.1182/bloodadvances.2018024026 (2019).
- 15 Starr, M. E. *et al.* A New Cecal Slurry Preparation Protocol with Improved Long-Term Reproducibility for Animal Models of Sepsis. *PloS one* **9**, e115705, doi:10.1371/journal.pone.0115705 (2014).

Reviewers' Comments:

Reviewer #1:

Remarks to the Author:

the authors have very nicely addressed my concerns in the revision.

Reviewer #2:

Remarks to the Author:

The authors have addressed all my questions and concerns. I have no further questions.

Reviewer #3:

Remarks to the Author:

All previous concerns have been addressed.

Reviewer #4:

Remarks to the Author:

Authors revised MS appropriately.

A point-by-point response to the reviewers' comments, final revisions for Nature Communications manuscript NCOMMS-20-34509

REVIEWERS' COMMENTS

Reviewer #1 (Remarks to the Author):

the authors have very nicely addressed my concerns in the revision.

Reviewer #2 (Remarks to the Author):

The authors have addressed all my questions and concerns. I have no further questions.

Reviewer #3 (Remarks to the Author):

All previous concerns have been addressed.

Reviewer #4 (Remarks to the Author):

Authors revised MS appropriately.

Response: We thank the reviewer for this positive feedback and their help in improving the paper. Since no additional changes were requested we have no additional comments.